# AMORTIZED SHAP VALUES VIA SPARSE FOURIER FUNCTION APPROXIMATION

## ABSTRACT

SHAP values – a.k.a. SHapley Additive exPlanations – are a popular local feature-attribution method widely used in interpretable and explainable AI. We tackle the problem of efficiently computing these values. We cover both the model-agnostic (black-box) setting, where one only has query access to the model and also the case of (ensembles of) trees where one has access to the structure of the tree. For both the black-box and the tree setting we propose a two-stage approach for estimating SHAP values.

Our algorithm's first step harnesses recent results showing that many real-world predictors have a spectral bias that allows us to either *exactly represent* (in the case of ensembles of decision trees), or *efficiently approximate* them (in the case of neural networks) using a compact Fourier representation. For the case of trees, given access to the tree structure, one can extract the Fourier representation using a simple recursive algorithm. For the black-box setting, given query access to the black-box function, we utilize a sparse Fourier approximation algorithm to efficiently extract its compact Fourier approximation.

In the second step of the algorithm, we use the Fourier representation to *exactly* compute SHAP values. The second step is computationally very cheap because firstly, the representation is compact and secondly, we prove that there exists a closed-form expression for SHAP values for the Fourier basis functions. Furthermore, the expression we derive effectively "linearizes" the computation into a simple summation and is amenable to parallelization on multiple cores on a GPU. Since the function approximation (first step) is only done once, it allows us to produce Shapley values in an *amortized* way. We show speedups compared to relevant baseline methods equal levels of accuracy for both the tree and black-box settings. Moreover, this approach introduces a reliable and fine-grained continuous trade-off between computation and accuracy through the sparsity of the Fourier approximation, a feature previously unavailable in all black-box methods.

## 1 INTRODUCTION

Interpretability of machine learning models is paramount, especially in high-stakes applications in areas such as medicine, fraud detection, or credit scoring. This is crucial to the extent that in Europe, the General Data Protection Regulation (GDPR) mandates the *legal* right to an explanation of algorithmic decisions (Voigt & Von dem Bussche, 2017). Say we are given a predictor/model $h : \mathcal{X}^n \to \mathbb{R}$ which maps an input (data) instance $x^* \in \mathcal{X}^n$ to a prediction $h(x^*)$. *Instance-wise* a.k.a. *local* feature attribution methods assign "importances" to each of the features $x_i^* \in \mathcal{X}$ of the instance $x^*$ which quantify how influential that feature was in the model predicting the value $h(x^*)$.

A widely used method for deriving attributions (importances) is the notion of *SHapley Additive exPlanations*, commonly referred to simply as SHAP values. Originally, the notion of Shapley values was introduced in the seminal work of Shapley (1952) in the context of cooperative game theory. The Shapley value is a mathematically well-founded and "fair" way of distributing a reward among all the members of a group playing a cooperative game and it is computed based on the rewards that would be received for all possible coalitions. The Shapley value is the unique way of distributing the reward that satisfies several reasonable mathematical properties that capture a notion of fairness (Shapley, 1952). In the context of machine learning and statistics, the players become features, the

reward is the prediction of the predictor $h$ and the SHAP value is the "contribution" or "influence" of that feature on the prediction. Shapley values are widely used due to their mathematical soundness and desirable properties Gromping (2007); Štrumbelj et al. (2009); Owen (2014); Datta et al. (2016); Owen & Prieur (2017); Lundberg & Lee (2017); Lundberg et al. (2020) Aas et al. (2021).

Despite their prevalence, computing SHAP values is challenging, as it involves an exponential summation, i.e. a summation over exponentially many terms , see Equation 1. This is because the formula accounts for a feature's importance in the context of all possible "coalitions" of other features, and therefore the formula covers all possible subsets of other features. Therefore, approximating them and speeding up the computation has received attention in a variety of settings. SHAP value computation can easily dominate the computation time of industry-level machine learning solutions on datasets with millions or more entries (Yang, 2021). Yang (2021) point out that industrial applications sometimes require hundreds of millions of samples to be explained. Examples include feed ranking, ads targeting, and subscription propensity models. In these modeling pipelines, spending tens of hours in model interpretation becomes a significant bottleneck (Yang, 2021) and one usually needs to resort to multiple cores and parallel computing.

Significant work has gone into speeding up the computation of SHAP values for a variety of settings. In the (ensemble of) trees setting, full access to the tree structure is assumed. Yang (2021); Bifet et al. (2022) provide theoretical and practical computational speedups to the classic TREESHAP (Lundberg et al., 2020). Similar to these results, in this work, we provide significant speedups for the tree setting over previous methods.

As opposed to the tree setting, which is a "white box" setting, in the *model-agnostic* a.k.a *black-box* setting, we only have *query access* to the model. Here, our only means of access to the predictor is that we can pick an arbitrary $x \in \mathcal{X}^n$ and query the predictor for its value $h(x)$. The usual approach here is to approximate the exponential sum of the SHAP value computations using stochastic sampling (Covert & Lee, 2020; Mitchell et al., 2022a; Lundberg & Lee, 2017). In this setting, Covert & Lee (2020); Mitchell et al. (2022a) provide sampling methods that require fewer queries to the black-box compared to vanilla KERNELSHAP (Lundberg & Lee, 2017) for equal approximation accuracy. FASTSHAP Jethani et al. (2021), which introduces a method for estimating Shapley values in a single forward pass using an end-to-end learned explainer neural network model. Our algorithm FOURIERSHAP falls into the query-access black-box setting as well.

However, we take a different approach. We are guided by the key insight that many models used in practice have a "spectral bias". Yang & Salman (2019); Valle-Perez et al. (2018) provably and experimentally show that fully connected neural networks with binary (zero-one) inputs learn low-degree – and therefore sparse – functions in a basis called the *Walsh-Hadamard a.k.a Fourier* basis. It is well known that the Walsh-Hadamard transform (WHT) of an ensemble of $T$ trees of depth $d$ is also of degree at most $d$ and moreover, $k = O(T4^d)$-sparse (Kushilevitz & Mansour, 1993; Mansour, 1994).

**Our contributions:** Guided by the aforementioned insights, we provide an algorithm to compute SHAP values, using the Fourier representation of the tree or black-box model. We first approximate the black-box function by taking its sparse Fourier transform. We theoretically justify, and show through extensive experiments, that for many real-world models such as fully connected neural networks and (ensembles of) trees this representation is accurate. Subsequently, we prove that SHAP values for a single Fourier basis function admit a closed-form expression not involving an exponential summation. Therefore, using the Fourier representation we overcome the exponential sum and can utilize compute power effectively to compute SHAP values. Furthermore, the closed-form expression we derive effectively "linearizes" the computation into a simple summation and is amenable to parallelization on multiple cores or a GPU. The Fourier approximation step is only done *once*, therefore FOURIERSHAP amortizes the cost of computing explanations for many inputs. Subsequently, SHAP computations using the Fourier approximation are orders of magnitude faster compared to other black-box SHAP approximation methods such as KERNELSHAP and other variations of it which all involve a computationally expensive optimization. We show speedups over other methods such as DEEPLIFT (Shrikumar et al., 2017) and FASTSHAP (Jethani et al., 2021) as well. In addition to the speedup, our algorithm enables a reliable continuous trade-off between computation and accuracy, controlled in a fine manner by the sparsity of the Fourier approximation. This was not formerly possible with these two black box methods.

## 2 BACKGROUND

This section reviews the notion of SHAP values, and sparse and low-degree Fourier transforms.

### 2.1 SHAPLEY VALUES

In game theory, a *cooperative game* is a function $v : 2^{[n]} \to \mathbb{R}$ that maps a subset (coalition) $S \subseteq [n]$ of a group of players $[n] = \{1, \ldots, n\}$ to their total reward (when they cooperate). If all the players cooperate they win a total reward of value $v([n])$. The main question is how they would distribute this reward among themselves. Shapley (1952) resolved this problem by a deriving a *unique* value based on "fairness axioms" proposed in his seminal work (Shapley, 1952). The *Shapley value* of player $i \in [n]$ is:

$$\phi_i(v) = \frac{1}{n} \sum_{S \subseteq [n] \setminus \{i\}} \frac{v(S \cup \{i\}) - v(S)}{\binom{n-1}{|S|}} \tag{1}$$

Intuitively, one can view the term $v(S \cup \{i\}) - v(S)$ as the marginal contribution of player $i$ when they are added to the coalition $S$. This marginal value is weighted according to the number of permutations the leading $|S|$ players and trailing $n - |S| - 1$ players can form.

In the machine learning context, we have a predictor $h : \mathcal{X}^n \to \mathbb{R}$ mapping an $n$-dimensional feature vector to a value. In this context, the players become features $x_i \in \mathcal{X}$ and the reward is the prediction of the predictor $h$ and the Shapley value is the "contribution" or "influence" of the $i$'th feature on the prediction. We define $v(S)$ accordingly to capture this notion (Lundberg & Lee, 2017):

$$v(S) = \mathbb{E}_{\boldsymbol{x}_{[n] \setminus S} \sim p(\boldsymbol{x}_{[n] \setminus S})}[h(x_S^*, \boldsymbol{x}_{[n] \setminus S})],$$

where $x^* \in \mathcal{X}^n$ is the instance we are explaining. This definition implicitly captures the way we handle the missing features (feature not present in the coalition): we integrate the missing features with respect to the marginal distribution $p(\boldsymbol{x}_{[n] \setminus S})$. In practice, the marginalization is performed with an empirical distribution by taking a subset of the training data as *background dataset*.

The choice of which distribution to average the missing features from has been discussed thoroughly in the relevant literature. As mentioned before, in this work, we focus on the SHAP values as defined in KERNELSHAP introduced by Lundberg & Lee (2017); Lundberg et al. (2020) also known as "Interventional" (Janzing et al., 2020; Van den Broeck et al., 2021) or "Baseline" (Sundararajan & Najmi, 2020) SHAP, where the missing features are integrated out from the *marginal distribution* $p(\boldsymbol{x}_{[n] \setminus S})$, as opposed to the *conditional distribution* $p(\boldsymbol{x}_{[n] \setminus S} | \boldsymbol{x}_S)$. We refer the reader to Appendix B.3 for a comprehensive overview of the literature discussing these two notions and their conceptual differences.

Since we will be using the well-known KERNELSHAP (Lundberg & Lee, 2017) and its variant LINREGSHAP (Covert & Lee, 2020) as a baseline we briefly review their method here. Lundberg & Lee (2017) propose the "least squares characterization" of SHAP values. They prove that SHAP values are the solution to the following minimization problem:

$$\beta_0^*, \ldots, \beta_n^* \triangleq \arg \min_{\beta_0, \ldots, \beta_n} \sum_{0 < |S| < n} \frac{n-1}{\binom{n}{|S|}|S|(n-|S|)} \left( \beta_0 + \sum_{i \in S} \beta_i - v(S) \right)$$

$$s.t. : \quad \beta_0 = v(\{\}), \beta_0 + \sum_{i=1}^n \beta_i = v([n])$$

Then $\phi_i(v) = \beta_i^*$.

Unfortunately, the above optimization still involves an exponential sum. Therefore, Lundberg & Lee (2017) propose to sample subsets $S$ uniformly at random. Covert & Lee (2020); Mitchell et al. (2022a) provide better ways of sampling and solving the optimization to get approximations with lower variances and biases. Nevertheless, all these methods require solving a least squares minimization subject to constraints for each explained instance $x^*$ and, therefore, are computationally expensive.

Later, we also compare our method to FASTSHAP (Jethani et al., 2021), a model-agnostic algorithm for computing SHAP values. In FASTSHAP, a parametric explainer $\phi$ (e.g., MLP) is trained to directly generate SHAP values given data samples. Since training is only done once, this, similar to us, amortizes the cost of generating SHAP values across multiple instances. Computing SHAP values only requires a forward pass on the trained model and therefore can be done very quickly.

## 2.2 FOURIER REPRESENTATIONS

Here we review the notions of the Fourier basis and what we mean by sparsity. We will later use sparse Fourier representation of functions to compute SHAP values. In this work, we focus on the setting where the inputs to the black-box function (predictor) are binary, i.e., $\mathcal{X}^n = \{0,1\}^n$. This means, we assume we have binary features and/or categorical features, through standard one-hot representations [1]. The Fourier representation of the *pseudo-boolean* function $h : \{0,1\}^n \to \mathbb{R}$ is the unique expansion of $h$ as follows: $h(x) = \frac{1}{\sqrt{2^n}} \sum_{f \in \{0,1\}^n} \widehat{h}(f)(-1)^{\langle f,x \rangle}$.

The inner product of two vectors $f, x \in \{0,1\}^n$ is defined as: $\langle f, x \rangle \equiv \sum_i^n f_i x_i, \forall f, x \in \{0,1\}^n$.

The unique function $\widehat{h} : \{0,1\}^n \to \mathbb{R}$ is called the *Fourier transform* of $h$. For any $f \in \{0,1\}^n$, $\widehat{h}(f)$ is called the Fourier coefficient corresponding to the frequency $f$. The family of functions $\frac{1}{\sqrt{2^n}}\Psi_f(x) = (-1)^{\langle f,x \rangle}, f \in \{0,1\}^n$ are the $2^n$-many Fourier basis functions. These basis functions are orthonormal: $\sum_{x \in \{0,1\}^n} \Psi_f(x)\Psi_{f'}(x) = \begin{cases} 0 & f \neq f' \\ 1 & f = f' \end{cases}$, $f, f' \in \{0,1\}^n$. Therefore, they form a basis for the vector space of all pseudo-boolean functions $h : \{0,1\}^n \to \mathbb{R}$.

We define the support of $h$ to be $\text{supp}(h) = \{f \in \{0,1\}^n | \widehat{h}(f) \neq 0\}$. We say that a function $h$ is $k$-*sparse* if at most $k$ of the $2^n$ Fourier coefficients $\widehat{h}(f)$ are non-zero, i.e., $|\text{supp}(h)| \leq k$. The degree of a vector $f \in \{0,1\}^n$ is denoted by $\deg(f)$ and is defined as the number of ones in the vector. For example, if $n = 5$ then $f = (1,0,0,1,0)$ is a vector of degree $\deg(f) = 2$. We say a function is *degree* $d$ when the frequencies $f \in \{0,1\}^n$ corresponding to non-zero Fourier coefficients are of degree less or equal to $d$ i.e. $\forall f \in \text{supp}(h)$ it holds that $\deg(f) \leq d$.

By definition of the Fourier basis, a $k$-sparse degree $d$ function can be written as a summation of $k$ (Fourier basis) functions, each one depending on at most $d$ input variables. The converse is also true:

**Proposition 1.** *Assume $h : \{0,1\}^n \to \mathbb{R}$ can be decomposed as follows: $h(x) = \sum_{i=1}^p h_i(x_{S_i}), S_i \subseteq [n]$. That is, each function $h_i : \{0,1\}^{|S_i|} \to \mathbb{R}$ depends on at most $|S_i|$ variables. Then, $h$ is $k = O(\sum_{i=1}^p 2^{|S_i|})$-sparse and of degree $d = \max(|S_1|, \ldots, |S_p|)$. (Proof in Appendix C.1)*

The sparsity $k$ and degree $d$ capture a notion of complexity for the underlying function. Intuitively speaking, the sparsity factor $k$ puts a limit on the number of functions in the decomposition, and the degree $d$ puts a limit on the order of interactions among the input variables.

One can see from Proposition 1, that modular functions, i.e., functions that can be written as a sum of functions each depending on exactly one variable, are $k = O(n)$-sparse and of degree $d = 1$. A slightly more "complex" function capturing second-order interactions among the input variables, i.e., a function that can be written as a sum where each term depends on at most two variables is going to be $k = O(n^2)$-sparse and of degree $d = 2$. The following proposition generalizes this result.

**Proposition 2.** *Let, $h : \{0,1\}^n \to \mathbb{R}$ be a pseudo-boolean function and let $d \in \mathbb{N}$ be some constant (w.r.t. $n$). If $h$ is of degree $d$, then it is $k = O(n^d)$-sparse. (Proof in Appendix C.1)*

This proposition formally shows that limiting the order of interactions among the input variables implies an upper bound on the sparsity.

---

[1] Continuous features can be discretized into quantiles, enabling their transformation into categorical features.

# 3 MANY REAL-WORLD BLACK-BOX PREDICTORS HAVE SPARSE FOURIER TRANSFORMS

In this section, we discuss the sparsity of the Fourier transforms of ensembles of trees and why neural networks can be approximated by a sparse Fourier representation because of their spectral bias. This shows both these classes of functions can be compactly represented in the Fourier basis. The results here will become useful in the next section, where we present our main contribution on how we can leverage this compact representation, to precisely compute SHAP values cheaply.

## 3.1 SPECTRAL BIAS OF FULLY CONNECTED NEURAL NETWORKS

The function a fully connected neural network represents at initialization is a sample from a Gaussian Process (GP) (Rasmussen, 2004) in the infinite-width limit. Here, the randomness is over the initial weights and biases. The kernel $K$ of this GP is called the Conjugate Kernel (CK) (Daniely et al., 2016; Lee et al., 2017). As mentioned before, here we investigate the case where the inputs to the neural network are binary, i.e., $\mathcal{X}^d = \{0,1\}^d$, similar to (Yang & Salman, 2019; Valle-Perez et al., 2018). The CK kernel Gram matrix formed on the whole input space $\mathcal{X}^d = \{0,1\}^d$, has a simple eigenvalue decomposition :

$$\mathcal{K} \in \mathbb{R}^{2^d \times 2^d}, \mathcal{K} = \sum_{f \in \{0,1\}^n} \lambda_f u_f u_f^\top$$

where $u_f \in \mathbb{R}^{2^d}$ is the eigenvector formed by evaluating the Fourier basis function $\Psi_f$ for different values of $x \in \{0,1\}^n$. Moreover, Yang & Salman (2019) show a weak spectral bias result in terms of the degree of $f$. Namely, the eigenvalues corresponding to higher degree frequencies have smaller values [2]. Given the kernel, $K$, and viewing a randomly initialized neural network function evaluated on the boolean $\{0,1\}^n$ as a sample from a GP one can see the following: This sample, roughly speaking, looks like a linear combination of the eigenvectors with the largest eigenvalues.

This is due to the fact that a sample of the GP can be obtained as $\sum_{i=1}^{2^n} \lambda_i \boldsymbol{w_i} u_i, \boldsymbol{w_i} \sim \mathcal{N}(0,1)$.

Combining this with the spectral bias results implies that neural networks are low-degree functions when randomly initialized.

Going beyond neural networks at initialization, numerous studies have investigated the behavior of fully connected neural networks trained through (stochastic) gradient descent. Chizat et al. (2019); Jacot et al. (2018b); Du et al. (2018); Allen-Zhu et al. (2019a;b) found that the weights of infinite-width neural networks after training do not deviate significantly from their initialization, which has been dubbed "lazy training" by Chizat et al. (2019). Lee et al. (2018; 2019) showed that training the last layer of an infinite-width randomly initialized neural network for an infinite amount of time corresponds to Gaussian process (GP) posterior inference with a certain kernel. Jacot et al. (2018b) extended the aforementioned results to training *all* the layers of a neural network (not just the final layer). They showed the evolution of an infinite-width neural network function can be described by the "Neural Tangent Kernel" (NTK, Jacot et al., 2018a) with the trained network yielding, on average, the posterior mean prediction of the corresponding GP after an infinite amount of training time. Lee et al. (2019) empirically showed the results carry over to the finite-width setting through extensive experiments. Yang & Salman (2019) again showed that the $u_f \in \mathbb{R}^{2^d}$ defined above are eigenvectors of the NTK Gram matrix and spectral bias holds. Gorji et al. (2023) validated these theoretical findings through extensive experiments in finite-width neural networks by showing that neural networks have "less tendency" to learn high-degree frequencies. We refer the reader to Appendix B.1 for a more comprehensive review.

The aforementioned literature shows that neural networks can be approximated by low-degree functions. By Proposition 2, they can be approximated by sparse functions for a large enough sparsity factor $k$. Our experiments in Section 5 provide further evidence that neural networks trained on real-world datasets are approximated well with sparse (and therefore compact) Fourier representations.

---

[2]To be more precise, they show that the eigenvalues corresponding to even and odd degree frequencies form decreasing sequences. That is, even and odd degrees are considered separately.

## 3.2 Sparsity of ensembles of decision trees

In our context, a decision tree is a rooted binary tree, where each non-leaf node corresponds to one of $n$ binary (zero-one) features, and each leaf node has a real number assigned to it. We denote the function a decision tree represents by $t : \{0, 1\}^n \to \mathbb{R}$. Let $i \in [n]$ denote the feature corresponding to the root, and let $t_{\text{left}} : \{0, 1\}^{n-1} \to \mathbb{R}$ and $t_{\text{right}} : \{0, 1\}^{n-1} \to \mathbb{R}$ be the left and right sub-trees, respectively. Then the tree can be represented as:

$$t(x) = \frac{1 + (-1)^{\langle e_i, x \rangle}}{2} t_{\text{left}}(x) + \frac{1 - (-1)^{\langle e_i, x \rangle}}{2} t_{\text{right}}(x) \tag{2}$$

where, $e_i \in \{0, 1\}^n$ is $i$'th indicator vector.

Thus, the Fourier transform of a decision tree can be computed recursively (Kushilevitz & Mansour, 1993; Mansour, 1994). The degree of a decision tree function of depth $d$ is $d$, and if $|\text{supp}(t_{\text{left}})| = k_{\text{left}}$ and $|\text{supp}(t_{\text{right}})| = k_{\text{right}}$, then $|\text{supp}(t)| \leq 2(k_{\text{left}} + k_{\text{right}})$. As a result, a decision tree function is $k$-sparse, where $k = O(4^d)$, although in some cases, when the decision tree is not balanced or cancellations occur, the Fourier transform can be sparser, i.e., admit a lower $k$, than the above upper bound on the sparsity suggests.

Due to the linearity of the Fourier transform, the Fourier transform of an *ensemble of trees*, such as those produced by the random forest, cat-boost (Dorogush et al., 2018), and XGBoost (Chen & Guestrin, 2016) algorithms/libraries, can be computed by taking the average of the Fourier transform of each tree. If the random forest model has $T$ trees, then its Fourier transform is $k = O(T4^d)$-sparse and of degree $d$ equal to its maximum depth of the constituent trees.

# 4 Computing SHAP values with Fourier representation of functions

In the previous section, we saw that many real-world models trained on tabular/discrete data can be exactly represented (in the case of ensembles of decision trees), or efficiently approximated (in the case of neural networks) using a compact (sparse) Fourier representation. We saw neural networks have a tendency to learn approximately low-degree, and hence by Proposition 2 sparse, functions. This has been attributed in numerous works to the reason why they generalize well and do not overfit despite their over-parameterized nature (Valle-Perez et al., 2018; Yang & Salman, 2019; Huh et al., 2022; Durvasula & Liter, 2020; Kalimeris et al., 2019; Neyshabur et al., 2017; Arpit et al., 2017). We also saw that (ensembles) of decision trees, by nature, have sparse Fourier representations (Kushilevitz & Mansour, 1993; Mansour, 1994). More generally, as made formal in Proposition 1 and the remarks after, any "simple" function that can be written as a summation of a "few" functions each depending on a "few" of the input variables is sparse and low-degree in the Fourier domain.

We propose the following method to approximate SHAP values for black-box functions. In the first step, given query access to a black-box function, we utilize a sparse Fourier approximation algorithm such as (Li & Ramchandran, 2015; Amrollahi et al., 2019; Li et al., 2015) to efficiently extract its sparse and hence compactly represented Fourier approximation. See Appendix B.2 for a more detailed explanation. Next, in our second step presented here, we use the Fourier representation to exactly compute SHAP values.

Let $h : \{0, 1\}^n \to \mathbb{R}$ be some predictor which we assume to be $k$-sparse with $n$ binary input features. Let $[n] \triangleq \{1, \ldots, n\}$ be the set of features and let $x^* \in \{0, 1\}^n$ be the instance we are explaining. As outlined in Equation 1, SHAP values are obtained from the following equation:

$$\phi_i^h = \sum_{S \subseteq [n] \setminus \{i\}} \frac{|S|!(n - |S| - 1)!}{M!} (v_h(S \cup \{i\}) - v_h(S)), \forall i \in [n] = \{1, 2, \ldots, n\}$$

where $v_h(S)$ is the average prediction when one only knows $x_S^*$. The average is taken over the (background) dataset $\mathcal{D} = \{(x, y)^i\}$ (Lundberg & Lee, 2017). More precisely:

$$v_h(S) \triangleq \frac{1}{|\mathcal{D}|} \sum_{(x,y) \in \mathcal{D}} h(x_S^*, x_{[n] \setminus S})$$

The above equations show the SHAP values are linear with respect to the prediction function $h$. Therefore we proceed by computing the Shapley values for a single Fourier basis function $\Psi_f(x) = (-1)^{\langle f,x \rangle}, f \in \{0,1\}^n$:

$$\phi_i^{\Psi_f} = \frac{1}{|\mathcal{D}|} \sum_{S \subseteq [n] \setminus \{i\}} \frac{|S|!(n-|S|-1)!}{n!} \cdot \sum_{(x,y) \in \mathcal{D}} \left( (-1)^{\langle f, x_{S \cup \{i\}}^* \oplus x_{[n] \setminus S \cup \{i\}} \rangle} - (-1)^{\langle f, x_S^* \oplus x_{[n] \setminus S} \rangle} \right)$$

(3)

The $\oplus$ operator concatenates two vectors along the same axis.

This expression still has an exponential (in $n$) sum, since we are summing over all subsets $S$. As a main contribution, we find a closed-form analytic expression for the inner summation using a combinatorial argument. This results in the following key Lemma:

**Lemma 1.** *Let $\Psi_f(x) = (-1)^{\langle f,x \rangle}$ be the Fourier basis function for some $f \in \{0,1\}^n$. Then the SHAP value of the Fourier basis function $\Psi_f$ with respect to the background dataset $\mathcal{D}$ is given as:*

$$\phi_i^{\Psi_f} = -\frac{2f_i}{|\mathcal{D}|} \sum_{(x,y) \in \mathcal{D}} \mathbb{1}_{x_i \neq x_i^*} (-1)^{\langle f,x \rangle} \frac{(|A|+1) \mod 2}{|A|+1}$$

(4)

*where $A \triangleq \{j \in [n] | x_j \neq x_j^*, j \neq i, f_j = 1\}$. (Proof in Appendix C.2)*

Finally, by the linearity of SHAP values w.r.t. the explained function $h$, and by utilizing Lemma 1 we arrive at the final expression for SHAP values of $h$. We present this closed-form expression alongside its computational complexity in our main Theorem:

**Theorem 2.** *Let $h : \{0,1\}^n \to \mathbb{R}$ be a $k$-sparse pseudo-boolean function with Fourier frequencies $f^1, \ldots, f^k \in supp(h)$ and amplitudes $\widehat{h}(f), \forall f \in supp(f)$. Let $\mathcal{D}$ be a background dataset of size $|\mathcal{D}|$. Then, Equation 5 provides a precise expression for the SHAP value vector $\phi^h = (\phi_1^h, \ldots, \phi_n^h)$. One can compute this vector with $\Theta(n \cdot |\mathcal{D}| \cdot k)$ flops (floating point operations).*

$$\phi_i^h = -\frac{2}{|\mathcal{D}|} \sum_{f \in supp(h)} \widehat{h}(f) \cdot f_i \cdot \sum_{(x,y) \in \mathcal{D}} \mathbb{1}_{x_i \neq x_i^*} (-1)^{\langle f,x \rangle} \frac{(|A|+1) \mod 2}{|A|+1}$$

(5)

*where $A$ is the same as in Lemma 1. (Proof in Appendix C.3)*

Theorem 2 gives us a computationally efficient way to go from a Fourier representation of a function $h$ to SHAP values. The SHAP values from this equation are *exact*, i.e., as long as the Fourier representation is exact, the SHAP values are also precise values. This is in contrast to KERNELSHAP, where the SHAP values are approximated using stochastic sampling and one needs to check for convergence to make sure the approximation is accurate. The approximation in our method is constrained to the first step: computing the (approximate) sparse Fourier representation of the black-box.

Most importantly, the sum in Equation 5 is *tractable*. This is because we overcome the exponential sum involved in Equation 1 by analytically computing the sum, with a combinatorial argument, for a single Fourier basis function. We show the number of flops that are required to compute SHAP values in Theorem 2 to be asymptotically equal to $\Theta(n \cdot |\mathcal{D}| \cdot k)$. We note that the $|\mathcal{D}|$ and $k$ factors in the asymptotic computational complexity arise from the two summations present in Equation 5. Through this expression, we are able to "linearize" the computation of SHAP values to a summation over the Fourier coefficients and background dataset. This allows us to maximally utilize the parallelization on multiple cores and/or GPUs to speed up the computation significantly. Therefore, in the presence of multiple cores or a GPU, we can get a speedup equal to the level of parallelization, as each core or worker can compute one part of this summation.

We implement our algorithm called FOURIERSHAP using JAX (Bradbury et al., 2018), which allows for fast vectorized operations on GPUs. Each term inside the summations of Equation 5 can be implemented with simple vector operations. Furthermore, summations over the $k$ different frequencies in the support of $h$ and also background data points can both be efficiently implemented and parallelized using this library using its VMAP operator. We perform all upcoming experiments on a single GPU. Nevertheless, we believe faster computation can also be achieved by crafting dedicated code designed to efficiently compute Equation 5 on GPU.

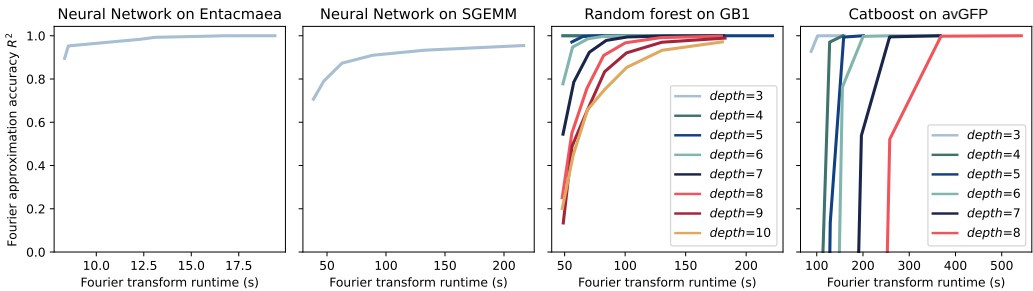

Figure 1: Step 1 of FOURIERSHAP: Accuracy of the Fourier transform (in approximating the black-box function) vs. runtime of the sparse Fourier algorithm. The accuracy is evaluated by $R^2$ score and comparing the outputs of the black-box and the Fourier representation on a uniformly generated random dataset on the Boolean cube $\{0, 1\}^n$. For a fixed level of accuracy, higher depth trees require a higher number of Fourier coefficients $k$ therefore a higher runtime. For the case of trees, we eventually are able to reach a perfect approximation since the underlying function is truly sparse.

Finally, we note that the function approximation (first step) is only done once, i.e., we compute the sparse Fourier approximation of the black-box only once. This is typically the most expensive part of the computation. For any new query to be explained, we resort to an efficient implementation of Equation 5. As our experiments will show, this yields orders of magnitudes faster computation than previous methods such as KERNELSHAP where, as mentioned before in Section 2, each explained instance requires solving an expensive optimization problem.

## 5 EXPERIMENTS

We assess the performance of our algorithm, FOURIERSHAP, on four different real-world datasets of varying nature and dimensionality. Three of our datasets are related to protein fitness landscapes (Poelwijk et al., 2019; Wu et al., 2016; Sarkisyan et al., 2016) and are referred to as "Entacmaea" (dimension $n = 13$), "GB1" ($n = 80$), and "avGFP" ($n = 236$) respectively. The fourth dataset is a GPU-tuning (Nugteren & Codreanu, 2015) dataset referred to as "SGEMM" ($n = 40$). The features of these datasets are binary (zero-one) and/or categorical with standard one-hot encodings. See Appendix D for dataset details.

For the Entacmaea and SGEMM datasets, we train fully connected neural networks with 3 hidden layers containing 300 neurons each. For GB1 we train ensembles of trees models of varying depths using the random forest algorithm and for avGFP we train again, ensembles of trees models of varying depths using the cat-boost algorithm/library (Dorogush et al., 2018).

**Black-box setting.** The first step of the FOURIERSHAP algorithm is to compute a sparse Fourier approximation of the black-box model. We use a GPU implementation of a sparse Walsh-Hadamard Transform (sparse WHT) a.k.a Fourier transform algorithm (Amrollahi et al., 2019) for each of the four trained models. The algorithm accepts a sparsity parameter $k$ which is the sparsity of the computed Fourier representation. Higher sparsity parameter $k$ results in a better function approximation but the sparse-WHT runtime increase linearly in $k$ as well. In Figure 1 we plot the accuracy of the Fourier function approximation as measured by the $R^2$-score for different values of $k$ (which result in different runtimes). The $R^2$ score is computed over a dataset formed by randomly sampling the Boolean cube $\{0, 1\}^n$.

The second step of FOURIERSHAP utilizes the Fourier approximation to compute SHAP values using Equation 5. We implement this step using JAX library (Bradbury et al., 2018), and run it on a GPU. For each model to be explained, we choose four different values for the number of background samples and four different values for the number of query points to be explained, resulting in a total of 16 runs of FOURIERSHAP for each model. Error bars capture these variations. We take the runtime of KERNELSHAP, with Github repo default settings, to be the base runtime all other methods are compared to, i.e., we assume its runtime is 1 unit.

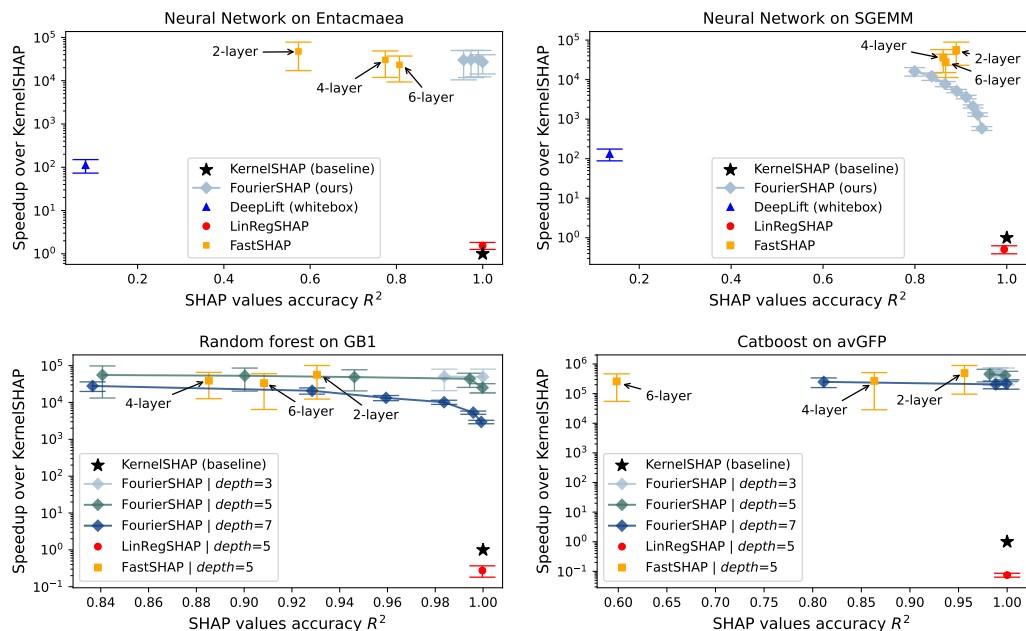

Figure 2: Speedup vs. Accuracy. Speedup of different algorithms is reported as a multiple compared to the runtime of KERNELSHAP. Accuracy is quantified by the $R^2$-score against ground truth SHAP values. DEEPLIFT(Shrikumar et al., 2017): is a white-box algorithm for neural networks. LINREGSHAP(Covert & Lee, 2020) is black-box algorithm and a variance-reduced version of KERNELSHAP. FASTSHAP(Jethani et al., 2021), a black box algorithm, is a trained MLP to output SHAP values given inputs in one forward pass. We experiment three sizes for MLPs for each dataset. FOURIERSHAP is ours. We are 10-10000x faster than LINREGSHAP on all dataset/model variations. More notably, we outperform DeepLift (a white-box algorithm) in the neural network model setting even though we assume only query access (black-box setting) to the neural network. We achieve higher accuracy than FASTSHAP in 3/4 settings, while enabling a fine-grained control over the speed-accuracy trade-off.

In order to measure the accuracy of the SHAP values produced by our and other algorithms we need ground truth SHAP values. We use KERNELSHAP to this extent. Note that KERNELSHAP is inherently an *approximation* method for computing interventional SHAP values. However, this approximation becomes more precise by sampling more coalition subsets. Therefore, to generate ground truth values, for each dataset, we sample more and more coalition subsets and check for convergence in these values. A more detailed explanation can be found in Appendix F.3.

We compute the $R^2$ values of Shapley values computed by FOURIERSHAP (ours) vs. ground truth values to evaluate accuracy. For our method, a higher sparsity $k$ for the Fourier representation results in a more accurate function approximation therefore higher $R^2$ values for the SHAP value quality. On the other hand, a higher $k$ results in a slower runtime as Equation 5 is a sum over the $k$ different frequencies. In Figure 2 we plot this trade-off.

We compare against the following baselines in Figure 2. The first is LINREGSHAP, a variance-reduced version of KERNELSHAP (Covert & Lee, 2020). We found that although this algorithm requires fewer queries from the black-box, it takes orders of magnitudes longer to run compared to ours. Secondly, for the neural network models, we also compare against a state-of-the-art *white-box* method – DEEPLIFT (Shrikumar et al., 2017). This algorithm, requires access to the neural network's activations in all layers. In comparison, we achieve a 10-100x speedup while being both more accurate and only assuming query access to the neural net (true black-box setting).

Finally, we compare against FASTSHAP (Jethani et al., 2021), which is the closest to our setting, i.e., the model agnostic black-box, amortized setting. We provide higher accuracy for similar values

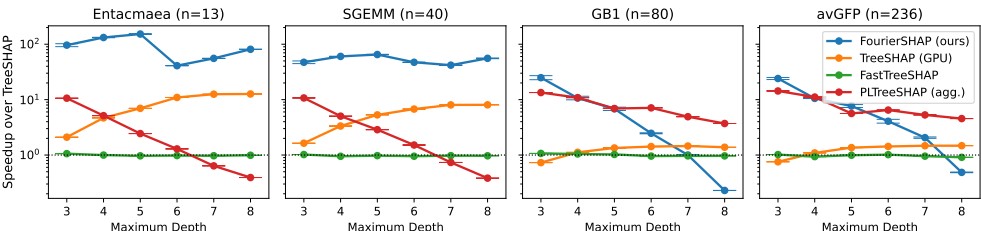

Figure 3: Speedup vs. depth of tree, for different algorithms, reported as a multiple compared to the runtime of TREESHAP Lundberg et al. (2020). **FOURIERSHAP** is ours. As other baselines we have a GPU implementation of **TREESHAP** (Mitchell et al., 2022b), **FASTTREESHAP** (Yang, 2021), and **PLTREESHAP** (Zern et al., 2023). We achieve order of magnitude speedups over all depths on the Entacmaea and SGEMM datasets. We also achieve significant speedups in the other two datasets; however, the edge diminishes as the maximum depth increases.

of speedup compared to FASTSHAP in 3/4 settings. More importantly though, we can see by controlling the sparsity parameter $k$ of the Fourier function approximation, we can control the tradeoff between accuracy and speed in a reliable and fine-grained manner. Whereas for FASTSHAP the accuracy of the SHAP values relies on the functional approximation properties of the MLP which directly produces SHAP values. As seen in Figure 2, the MLP can behave in unpredictable ways. In this figure we can see that increasing the depth of the model (and hence the approximation capability of the MLP) does not have a reliable effect on SHAP accuracy. Finally, in FASTSHAP for a different choice of background dataset a new MLP model has to be trained from scratch. Since, the MLP approximates the process of directly computing SHAP values. Whereas we can support different sets of background datasets since our Fourier functional approximation is done on the black box predictor, and not the whole process of producing SHAP values in an end-to-end manner (FASTSHAP).

**Tree setting.** FOURIERSHAP can also be utilized for the computation of SHAP values for (ensembles of) trees in the white-box setting, i.e., where full access to the tree's structure is available. In this setting, the exact sparse Fourier representation of an ensemble of trees can be efficiently computed (the first step of FOURIERSHAP) using Equation 2. With the exact Fourier representation at hand, SHAP values can be *efficiently and exactly* computed using Equation 5, the second step of FOURIERSHAP. This makes our method a highly parallelizable alternative for TREESHAP (Lundberg et al., 2020) with the potential for order of magnitude speedups.

To demonstrate our method's ability in fast computation of SHAP values in this setting, we compute SHAP values for random forests fitted on all aforementioned real-world datasets. We compare FOURIERSHAP's speedup over TREESHAP (Lundberg et al., 2020), to FASTTREESHAP (Yang, 2021), a fast implementation of TREESHAP, the GPU implementation of TREESHAP (Mitchell et al., 2022b) and PLTREESHAP (Zern et al., 2023). To the best of our knowledge, these are the fastest available frameworks for computation of the "interventional" SHAP values. Figure 3 shows the superior speed of our method over all state-of-the-art algorithms in most settings, which weakens by the increment of depth and number of features, resulting in more frequencies and computation overhead. Note that the SHAP values computed by all methods are precise and identical to the values produced by TREESHAP, which is to be expected for FOURIERSHAP, given access to exact Fourier representation of random forests.

## CONCLUSIONS

We illustrated in theory and practice how many black-box functions can be represented or efficiently approximated by a compact Fourier representation. We proved that SHAP values of Fourier basis functions admit a closed form expression, and therefore, we can compute SHAP values efficiently using the compact Fourier representation. Moreover, this closed form expression is amenable to parallelization. These two factors helped us in gaining speedups of 10-10000x over baseline methods for the computation of SHAP values.

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

# A APPENDIX

# B RELEVANT WORK

## B.1 NEURAL NETWORK THEORY AND SIMPLICITY BIASES

With regards to simplicity biases in fully connected neural networks, a substantial amount of research has been dedicated to analyzing neural networks in function space. This line of research is dedicated to firstly showing that "infinite-width", randomly initialized (with Gaussian distribution) neural networks are distributed as Gaussian Processes (GPs) and secondly computing the kernel associated to the GP (Neal, 1996; Williams, 1996; Cho & Saul, 2009; Hazan & Jaakkola, 2015; Lee et al., 2017; Ancona et al., 2018; Daniely, 2017). The kernel associated with the GP is commonly known as the "conjugate kernel" (Daniely, 2017) or the "NN-GP kernel"(Lee et al., 2017). Other works show that in infinite-width neural networks weights after training via SGD do not end up too far from the original (Chizat et al., 2019; Jacot et al., 2018a; Du et al., 2018; Allen-Zhu et al., 2019a;b), referred to as "lazy training" by Chizat et al. (2019). This allowed Jacot et al. (2018a) to prove that the evolution of an infinite-width neural network during training can be described by a kernel called the "Neural Tangent Kernel". Lee et al. (2019) showed, through extensive experiments that the same behavior holds even for the more realistic case of neural nets of finite width. Empirically speaking, it was shown by Rahaman et al. (2019) that a neural net with one input tends to learn sinusoids of lower frequencies in earlier epochs than those with higher frequencies. By analyzing the spectrum of the NTK's Gram matrix, Ronen et al. (2019); Basri et al. (2020) were able to formally prove this empirical finding. Yang & Salman (2019); Fan & Wang (2020) analyze the spectra of the NTK gram matrix for higher dimensional inputs. Specifically, Yang & Salman (2019); Valle-Perez et al. (2018) provide simplicity bias results for the case where the inputs to the neural net are Boolean (zero-one) vectors.

## B.2 SPARSE AND LOW-DEGREE FOURIER TRANSFORM ALGORITHMS

We now discuss algorithms that efficiently approximate *general* black-box predictors by a Fourier sparse representation. Let $h : \{0,1\}^n \to \mathbb{R}$ be a any function. We assume we have query access to $h$. That is, we can arbitrarily pick $x \in \{0,1\}^n$ and query $h$ for its value $h(x)$. Without any further assumptions, computing the Fourier transform requires us to query *exponentially*, to be precise $2^n$, many queries: one for every $x \in \{0,1\}^n$. Furthermore, classical Fast Fourier Transform (FTT) algorithms are known to take at least $\Omega(2^n \log(2^n))$ time.

Under the additional assumption that $h$ is $k$-sparse, works such as Cheraghchi & Indyk (2017); Amrollahi et al. (2019); Scheibler et al. (2015); Kushilevitz & Mansour (1993); Li & Ramchandran (2015); Li et al. (2015) provide algorithms that obtain the Fourier transform more efficiently. In particular, Amrollahi et al. (2019) provide algorithms with query complexity $O(nk)$ and time complexity $O(nk \log k)$ time. Assuming further that the function is of degree $d = o(n)$, the query complexity reduces to $O(kd \log n)$, with run time still polynomial in $n, k, d$. Crucially, even if the function $h$ is not $k$-sparse, Algorithm ROBUSTSWHT of Amrollahi et al. (2019) yields the best $k$-sparse approximation in the $\ell_2 - \ell_2$ sense. More precisely, let us denote by $h_k : \{0,1\}^n \to$ the function that is formed by only keeping the top $k$ non-zero Fourier coefficients of $h$ and setting the rest to zero. Then the algorithm returns a $O(k)$-sparse function $g$ such that:

$$\sum_{f \in \{0,1\}^n} (\widehat{g}(f) - \widehat{h}(f))^2 \le C(1+\epsilon) \min_{\text{all } k\text{-sparse } g} \sum_{f \in \{0,1\}^n} (\widehat{g}(f) - \widehat{h_k}(f))^2,$$

where $C$ is a universal constant. By Parseval's identity, the same holds if the summations were over the time (input) domain instead of the frequency domain.

## B.3 SHAPLEY VALUES IN THE CONTEXT OF MACHINE LEARNING

In the context of ML, many works have derived a different notion of Shapley value depending on what they mean by data distribution, deleted features, etc. We refer the reader to the survey by Sundararajan & Najmi (2020); Janzing et al. (2020) for a comprehensive overview. In this work we focus on the notion of SHAP introduced by Lundberg & Lee (2017); Lundberg et al. (2020) also

known as "Interventional" (Janzing et al., 2020; Van den Broeck et al., 2021) or "Baseline" SHAP (Sundararajan & Najmi, 2020) where the missing features are integrated out from the marginal distribution as opposed to the conditional distribution (see Section 2). As pointed out by Janzing et al. (2020) there are two main ways to define the SHAP value "interventional" and "observational" SHAP. These are referred to by Sundararajan & Najmi (2020) as "baseline" and "conditional" SHAP respectively.

As pointed out by Janzing et al. (2020) the difference between these definitions can be better viewed with the lens of causality (Pearl, 2009). Roughly speaking "observational" SHAP tells us about how influential a feature is to the prediction of the predictor if it goes from the state of being unobserved to observed. "Interventional" SHAP is *causal* and tells us how influential a feature is if we were to reach in (through a process called an intervention) and change that feature in order to change the prediction. Put into the context of credit scores and loan approvals, "observational" SHAP will provide us with important features which are "observed" by the predictor and hence are influential in predicting if a particular loan request will be rejected or approved. Interventional SHAP would tell us which feature we could change or "intervene" in order to change the outcome of the loan request.

The original (ML) SHAP paper (Lundberg & Lee, 2017) proposes "observational" SHAP as the correct notion of SHAP. Van den Broeck et al. (2021); Arenas et al. (2021) provide intractability results for observational SHAP in a variety of simple distributional assumptions on the data and simple predictors $f$. This has lead to many attempts to *approximate* observational SHAP values (Lundberg et al., 2020; Covert & Lee, 2020; Aas et al., 2021; Kwon et al., 2021; Sundararajan & Najmi, 2020). It is interesting to note that the version of "Kernel"-SHAP in Lundberg & Lee (2017) is also an approximation for observational SHAP values that ends up coinciding precisely with interventional SHAP values, which explains a lot of the confusion in the community. Janzing et al. (2020) boldly claims that researchers should stop the pursuit of approximations to "observation" SHAP values as it lacks certain properties, for example, sensitivity i.e. the SHAP value of a feature can be non-zero while the predictor $f$ has no dependence on that feature. This phenomenon happens because when features are correlated, the presence of a feature can provide information about other features that the predictor does depend on. This does *not* happen in interventional SHAP. Finally, Chen et al. (2020) argues that both SHAP definitions are worthy of pursuit. They emphasize that the interventional framework provides explanations that are more "true to the model", and the observational approach's explanations are more "true to the data".

## C PROOFS

Before we start with the proofs we review the Fourier analysis and synthesis equations. As we mentioned in the Background Section 2, the Fourier representation of the *pseudo-boolean* function $h : \{0,1\}^n \to \mathbb{R}$ is the unique expansion of $h$ as follows:

$$h(x) = \frac{1}{\sqrt{2^n}} \sum_{f \in \{0,1\}^n} \widehat{h}(f)(-1)^{\langle f, x \rangle}$$

This is the so called Fourier "synthesis" equation.

The Fourier coefficients $\widehat{h}(f)$ are computed by the Fourier "analysis" equation:

$$\widehat{h}(f) = \frac{1}{\sqrt{2^n}} \sum_{x \in \{0,1\}^n} h(x)(-1)^{\langle f, x \rangle} \tag{6}$$

### C.1 PROOF OF PROPOSITIONS

We can now prove Proposition 1:

**Proposition 1.** *Assume $h : \{0,1\}^n \to \mathbb{R}$ can be decomposed as follows: $h(x) = \sum_{i=1}^{p} h_i(x_{S_i})$, $S_i \subseteq [n]$. That is, each function $h_i : \{0,1\}^{|S_i|} \to \mathbb{R}$ depends on at most $|S_i|$ variables. Then, $h$ is $k = O(\sum_{i=1}^{p} 2^{|S_i|})$-sparse and of degree $d = \max(|S_1|, \ldots, |S_p|)$. (Proof in Appendix C.1)*

*Proof.* Let $g : \{0,1\}^n \to \mathbb{R}$ be a function dependent on exactly $d$ variables $x_{i_1}, \ldots, x_{i_d}$, where $i_1, \ldots, i_d \in [n]$ are distinct indices. We show that for any frequency $f \in \{0,1\}^n$, if $f_j = 1$ for some $j \notin S \triangleq \{i_1, \ldots, i_d\}$, then, $\widehat{g}(f) = 0$. From the Fourier analysis Equation 6 we have:

$$\widehat{g}(f) = \frac{1}{\sqrt{2^n}} \sum_{x \in \{0,1\}^n} g(x)(-1)^{\langle f, x \rangle} = \sum_{x_i : i \in S} \sum_{x_j : j \in [n] \backslash S} g(x)(-1)^{\langle f_S, x_S \rangle}(-1)^{\langle f_{[n] \backslash S}, x_{[n] \backslash S} \rangle}$$

$$\overset{(i)}{=} \sum_{x_i : i \in S} g(x)(-1)^{\langle f_S, x_S \rangle} \sum_{x_j : j \in [n] \backslash S} (-1)^{\langle f_{[n] \backslash S}, x_{[n] \backslash S} \rangle} \overset{(ii)}{=} \sum_{x_i : i \in S} g(x)(-1)^{\langle f_S, x_S \rangle} \cdot 0 = 0$$

Where Equation i holds because $g$ is only dependent on the variables in $S$ and Equation ii holds by checking the inner sum has an equal number of $1$ and $-1$ added together.

The proof of the proposition follows by the linearity of the Fourier transform. $\qquad\square$

Moving on to Proposition 2:

**Proposition 2.** *Let,* $h : \{0,1\}^n \to \mathbb{R}$ *be a pseudo-boolean function and let* $d \in \mathbb{N}$ *be some constant (w.r.t. $n$). If $h$ is of degree $d$, then, it is $k = O(n^d)$-sparse. (Proof in Appendix C.1)*

*Proof.* We simply note that the number of frequencies $f \in \{0,1\}^n$ of degree at most $d$ is equal to $\sum_{i=0}^{d} \binom{n}{i}$. This sum is $O(n^d)$ for $d$ constant w.r.t $n$. $\qquad\square$

## C.2 PROOF OF LEMMA 1

*Proof.* We start from Equation 3:

$$\phi_i^{\Psi_f} = \sum_{S \subseteq [n] \backslash \{i\}} \frac{|S|!(n - |S| - 1)!}{n!} \cdot \frac{1}{|\mathcal{D}|} \sum_{(x,y) \in \mathcal{D}} \left( (-1)^{\langle f, x_{S \cup \{i\}}^* \oplus x_{[n] \backslash S \cup \{i\}} \rangle} - (-1)^{\langle f, x_S^* \oplus x_{[n] \backslash S} \rangle} \right)$$

$$= \frac{1}{|\mathcal{D}|} \sum_{(x,y) \in \mathcal{D}} \sum_{S \subseteq [n] \backslash \{i\}} \frac{|S|!(n - |S| - 1)!}{n!} (-1)^{\langle f_{-i}, x_S^* \oplus x_{[n] \backslash S \cup \{i\}} \rangle} \left( (-1)^{f_i x_i^*} - (-1)^{f_i x_i} \right)$$

By checking all 8 possible combinations of $x_i, x_i^*, f_i \in \{0,1\}$, one can see that $(-1)^{f_i x_i^*} - (-1)^{f_i x_i} = 2f_i(x_i - x_i^*)$. This is simply because the two exponents only differ when $f_i = 1$ and $x_i \neq x_i^*$.

To determine $(-1)^{\langle f_{-i}, x_S^* \oplus x_{[n] \backslash S \cup \{i\}} \rangle}$, we partition $[n] \backslash \{i\}$ into two subsets $A \triangleq \{j \in [n] | x_j \neq x_j^*, j \neq i, f_j = 1\}$ and $B \triangleq [n] \backslash A \cup \{i\}$. Doing this, we can factor out $(-1)^{\langle f_{-i}, x_{-i} \rangle}$ and determine the rest of the sign based on the number of indices in $S$ where $x$ and $x^*$ disagree and $f_i = 1$. This is equal to $|A \cap S|$:

$$\phi_i^{\Psi_f} = \frac{2f_i}{|\mathcal{D}|} \sum_{(x,y) \in \mathcal{D}} (x_i - x_i^*)(-1)^{\langle f_{-i}, x_{-i} \rangle} \sum_{S \subseteq [n] \backslash \{i\}} \frac{|S|!(n - |S| - 1)!}{n!} (-1)^{|A \cap S|}$$

$A$ and $B$ partition $[n] \backslash \{i\}$, therefore we split the inner sum as follows:

$$\phi_i^{\Psi_f} = \frac{2f_i}{|\mathcal{D}|} \sum_{(x,y) \in \mathcal{D}} (x_i - x_i^*)(-1)^{\langle f_{-i}, x_{-i} \rangle} \sum_{\tilde{B} \subseteq B} \sum_{\tilde{A} \subseteq A} \frac{(|\tilde{A}| + |\tilde{B}|)!(n - (|\tilde{A}| + |\tilde{B}|) - 1)!}{n!} (-1)^{|\tilde{A}|}$$

Since the inner expression only depends on the cardinalities of $\tilde{A}$ and $\tilde{B}$ we can recast the inner sum to be over numbers instead of subsets by counting the number of times each cardinality appears in the summation:

$$\phi_i^{\Psi_f} = \frac{2f_i}{|\mathcal{D}|} \sum_{(x,y) \in \mathcal{D}} (x_i - x_i^*)(-1)^{\langle f_{-i}, x_{-i} \rangle} \sum_{b=0}^{n - |A| - 1} \sum_{a=0}^{|A|} \binom{n - |A| - 1}{b} \binom{|A|}{a} \frac{(a + b)!(n - a - b - 1)!}{n!} (-1)^a$$

$$= \frac{2f_i}{|\mathcal{D}|} \sum_{(x,y) \in \mathcal{D}} (x_i - x_i^*)(-1)^{\langle f_{-i}, x_{-i} \rangle} \sum_{a=0}^{|A|} (-1)^a \sum_{b=0}^{n - |A| - 1} \frac{\binom{|A|}{a} \binom{n - |A| - 1}{b}}{n \binom{n-1}{a+b}} \qquad (7)$$

Now we find a closed-form expression for the innermost summation in the above Equation, which is a summation over $b$ where $a$ is fixed:

$$\sum_{b=0}^{n-|A|-1} \frac{\binom{|A|}{a}\binom{n-|A|-1}{b}}{n\binom{n-1}{a+b}} \overset{(i)}{=} \sum_{b=0}^{n-|A|-1} \frac{\binom{|A|}{a}\binom{n-|A|-1}{b}}{n\frac{\binom{n-1}{|A|}}{\binom{a+b}{a}\binom{n-a-b-1}{|A|-a}}\binom{|A|}{a}\binom{n-|A|-1}{b}}$$

$$= \frac{1}{n\binom{n-1}{|A|}} \sum_{b=0}^{n-|A|-1} \binom{a+b}{a}\binom{n-a-b-1}{|A|-a}$$

$$\overset{(ii)}{=} \frac{1}{n\binom{n-1}{|A|}} \binom{n}{|A|+1}$$

$$= \frac{1}{|A|+1} \tag{8}$$

In Equation i, we use the following identity: $\binom{n-1}{a+b}\binom{a+b}{a}\binom{n-a-b-1}{|A|-a} = \binom{n-1}{|A|}\binom{|A|}{a}\binom{n-|A|-1}{b}$. This can be checked algebraically by simply writing down each binomial term as factorials and doing the cancellations:

$$\binom{n-1}{a+b}\binom{a+b}{a}\binom{n-a-b-1}{|A|-a}$$

$$= \frac{(n-1)!}{(a+b)!(n-a-b-1)!} \cdot \frac{(a+b)!}{a!b!} \cdot \frac{(n-a-b-1)!}{(|A|-a)!(n-a-b-1-(|A|-a))!}$$

$$= \frac{(n-1)!}{a!b!(|A|-a)!(n-|A|-b-1)!} \cdot$$

$$= \frac{(n-1)!}{|A|!(n-|A|-1)!} \cdot \frac{|A|!}{a!(|A|-a)!} \cdot \frac{(n-|A|-1)!}{b!(n-|A|-1-b)!}$$

$$= \binom{n-1}{|A|}\binom{|A|}{a}\binom{n-|A|-1}{b}$$

In Equation ii, we use $\sum_{b=0}^{n-|A|-1} \binom{a+b}{a}\binom{n-a-b-1}{|A|-a} = \binom{n}{|A|+1}$ which holds because of the following double-counting argument. The term $\binom{n}{|A|+1}$ counts the number of ways to choose a subset of size $|A|+1$ from a set of $n$ elements. Imagine elements are numbered from 1 to n, and $\pi_1 < \pi_2 < ... < \pi_{|A|+1} \in [n]$ represent $|A|+1$ chosen elements. Let's condition on $\pi_{a+1} = (a+b+1)$ where possible values for $b$ can only be $0 \le b \le n-|A|-1$. This is due to the fact that $\pi_{a+1} < a+1$ implies $\pi_1, ..., \pi_a$ are chosen from less than $a$ elements, and similarly $\pi_{a+1} > n-|A|+a$ implies $\pi_{a+2}, ..., \pi_{|A|+1}$ ($|A|-a$ chosen elements) are chosen from less than $|A|-a$ elements, which are both impossible. Given the condition, $a$ elements numbered lower than $(a+b+1)$ and $|A|-a$ elements numbered larger than $(a+b+1)$ are also chosen. This is possible in $\binom{a+b}{a}\binom{n-a-b-1}{|A|-a}$ ways.

Therefore, keeping in mind that $a$ is fixed, $\sum_{b=0}^{n-|A|-1} \binom{a+b}{a}\binom{n-a-b-1}{|A|-a}$ should give the number of ways to choose a subset of size $|A|+1$ from a set of $n$ elements, through the new perspective.

Based on Equation 8, we see that the innermost summation in Equation 7 is only dependent on $|A|$. Thus, we rewrite Equation 7 as follows:

$$\phi_i^{\Psi_f} = \frac{2f_i}{|\mathcal{D}|} \sum_{(x,y)\in\mathcal{D}} (x_i - x_i^*)(-1)^{\langle f_{-i}, x_{-i}\rangle} \frac{(|A|+1) \mod 2}{|A|+1}$$

By absorbing the sign of $(x_i - x_i^*)$ into the $(-1)^{\langle f_{-i}, x_{-i}\rangle}$ term we arrive at Equation 4:

$$\phi_i^{\Psi_f} = -\frac{2f_i}{|\mathcal{D}|} \sum_{(x,y)\in\mathcal{D}} \mathbb{1}_{x_i \neq x_i^*}(-1)^{\langle f, x\rangle} \frac{(|A|+1) \mod 2}{|A|+1}$$

$\square$

### C.3 Proof of Theorem 2

*Proof.* Proof of Equation 5 simply follows from the fact that SHAP values are linear w.r.t. the explained function. Regarding the computational complexity we restate Equation 5

$$\phi_i^h = -\frac{2}{|\mathcal{D}|} \sum_{f \in \text{supp}(h)} \widehat{h}(f) \cdot f_i \sum_{(x,y) \in \mathcal{D}} \mathbb{1}_{x_i \neq x_i^*} (-1)^{\langle f, x \rangle} \frac{(|A| + 1) \mod 2}{|A| + 1}$$

where $A \triangleq \{j \in [n] | x_j \neq x_j^*, j \neq i, f_j = 1\}$.

We first do a pre-processing step for amortizing the cost of computing $|A|$: we compute $\tilde{A} \triangleq \{j \in [n] | x_j \neq x_j^*, f_j = 1\}$ which takes $\Theta(|\mathcal{D}|n)$ flops.

We assume we are computing the whole vector $\Phi^h = (\Phi_1^h, \ldots, \Phi_n^h)$, that is we are compute SHAP values for all $i \in [n]$ at the same time. Going back to the inner summation above, computing $A$ (and $|A|$) for different values of $i \in [n]$ is $\Theta(n)$ if we utilize the pre-computed $\tilde{A}$. The inner product $\langle f, x \rangle$ is not dependent on $i$ and is $\Theta(n)$ flops. Computing $\mathbb{1}_{x_i \neq x_i^*}$ for different values of $i \in [n]$ is $\Theta(n)$. Therefore, the inner expression of the summand takes $\Theta(n)$ flops for a fixed data-point $x \in \mathcal{D}$ and $f \in \text{supp}(h)$.

Computing the inner sum for any fixed frequency $f \in \text{supp}(h)$ is $\Theta(n|\mathcal{D}|)$, because we are summing over $|\mathcal{D}|$ vectors each of size $n$ (the vector which holds SHAP value for each $i \in [n]$). Moving on to the outer sum each evaluation of the inner sum is $\Theta(n|\mathcal{D}|)$ and it results in a vector of size $n$ (one element for each SHAP value). The multiplication of $\widehat{h}(f) \cdot f_i$ is $\Theta(n)$. Therefore the cost of the inner sum dominates i.e. $\Theta(n|\mathcal{D}|)$. Since we are summing over the whole support the total number of flops is: $\Theta(n|\mathcal{D}|k)$ where $k = |\text{supp}(h)|$. $\qquad\square$

## D Datasets

We list all the datasets used in the Experiments Section 5.

**Entacmaea quadricolor fluorescent protein. (Entacmaea)** Poelwijk et al. (2019) study the fluorescence brightness of all $2^{13}$ distinct variants of the Entacmaea quadricolor fluorescent protein, mutated at 13 different sites.

**GPU kernel performance (SGEMM).** Nugteren & Codreanu (2015) measures the running time of a matrix product using a parameterizable SGEMM GPU kernel, configured with different parameter combinations. The input has 14 categorical features. After one-hot encoding the dataset is 40-dimensional.

**Immunoglobulin-binding domain of protein G (GB1).** Wu et al. (2016) study the "fitness" of variants of protein GB1, that are mutated at four different sites. Fitness, in this work, is a quantitative measure of the stability and functionality of a protein variant. Given the 20 possible amino acids at each site, they report the fitness for $20^4 = 160,000$ possible variants, which we represent with one-hot encoded 80-dimensional binary vectors.

**Green fluorescent protein from Aequorea victoria (avGFP).** Sarkisyan et al. (2016) estimate the fluorescence brightness of random mutations over the green fluorescent protein sequence of Aequorea victoria (avGFP) at 236 amino acid sites. We transform the amino acid features into binary features indicating the absence or presence of a mutation at each amino acid site. This converts the original $54,024$ distinct amino acid sequences of length 236 into $49,089$ 236-dimensional binary data points.

## E FourierShap Implementation

We implemented four versions of our algorithm, i.e., Equation 5, using Google JAX library (Bradbury et al., 2018). JAX provides a flexible framework for developing high-performace functions

for vectorized computations. JAX enables automatic performance optimisation of algebraic computation as well as just-in-time (JIT) compilation of the vectorized functions for faster iterations at runtime.

The full code base is provided as supplementary material to the paper. We refer the reader there for more details. Here we give a high level overview of the four versions we implemented and experimented with:

- **Base**: In this version, frequencies are represented as $n$-dimensional binary vectors. A simple implenetation is provided for computing the SHAP values given a single frequency, a single background instance, and a single query instance. This is next extended to multiple frequencies, multiple background instances, and multiple query instances using multiple JAX `vmaps`.

- **Precompute**: This version is a modified version of the Base version. We take a closer look at Equation 5 and pre-compute terms dependent only on frequencies $f$ and background dataset points $x$ (not including terms dependent on the query point $x^*$). These are then loaded from memory at run-time. This version was faster than Base in all of our experiments, but inherently requires more memory; in our case, more GPU memory.

- **Sparse**: Here we utilize the sparsity of frequencies $f \in \{0, 1\}^n$ i.e. the fact that frequencies have mostly zero entries in practice. In this version, for each frequency $f$, we focus on positions $i$ where $f_i = 1$. Inspecting Equation 5, these are the only positions where $f$ can affect the final SHAP value vector.

- **Positional**: This version brings again builds on the previous and computes SHAP values separately for each coordinate $i \in [n]$. This adds one extra precomputation step to map frequencies to coordinates they affect, but enables mathematical simplifications as well as potential for extra vectorization. We observed that although this extra precomputation step could become time-consuming for large set of frequencies, it can result in a significantly faster SHAP value computation at run-time in the case of low-degree Fourier spectrums, compared to the Sparse version.

## F  EXPERIMENT DETAILS

The code for running the experiments and the implementations of all modules will be open-sourced once the double-blind review process is over. We run all experiments on a machine with one NVIDIA GeForce RTX 4090 GPU, on servers with Intel(R) Xeon(R) CPU E3-1284L v4 @ 2.90GHz, restricting the memory/RAM to 20 GB, which was managed with Slurm.

### F.1  BLACK-BOX

For the Entacmaea and SGEMM datasets, we train fully connected neural networks with 3 hidden layers containing 300 neurons each. The network is trained using the means-squared loss and ADAM optimizer with a learning rate of 0.01. For GB1 we train ensembles of trees models of varying depths using the random forest algorithm using the sklearn library (Pedregosa et al., 2011). For avGFP we train again, ensembles of trees models with 10 trees of varying depths using the cat-boost algorithm/library(Dorogush et al., 2018). All other setting are set to the default in both cases. Model accuracy for different depths are plotted in Figure 4.

For each dataset, we use all possible combination of number of background samples and query instances (instances to be explained) from $\{10, 20, 30, 40\}$, resulting in a total of 16 runs per dataset. Error bars in Figure 2 capture the variation in speedups.

### F.1.1  FOURIERSHAP

For the first step of the FOURIERSHAP, i.e., computing a sparse Fourier approximation of the black-box model, we use an implementation of a sparse Walsh-Hadamard Transform (sparse WHT) a.k.a Fourier transform algorithm (Amrollahi et al., 2019) for each of the four trained models. The implementation is done using Google JAX (Bradbury et al., 2018) library as part of this work, and is available in the project repository.

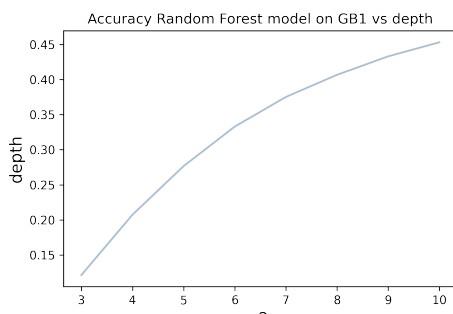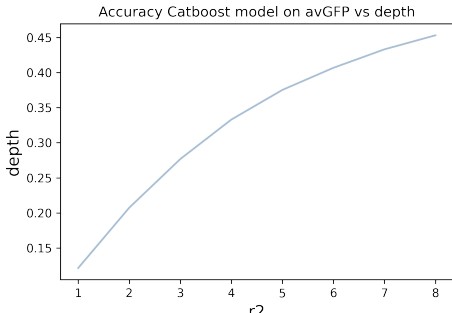

Figure 4: Model accuracy of tree models evaluated on the test set for different depths

For the second step of FOURIERSHAP, i.e., utilizing the approximated Fourier spectrum to compute SHAP values as per Equation 5, we use our "Precompute" implementation. We discussed the details in the previous Section, Section E.

To showcase the flexibility of our method in controlling the trade-off between speed and accuracy in Figure 2, we prune the computed sparse Fourier spectrum in the first step, and remove frequencies with amplitudes smaller than a specific threshold from the spectrum. For the threshold, we use 10 values between 0.0001 and 0.05 and specified a minimal descriptive subset of them as points in Figure 2.

### F.1.2 KERNELSHAP

For KERNELSHAP, we use the standard library provided by the writers of the paper (Lundberg & Lee, 2017) [3] with its default settings. As part of the default setting, the paired sampling trick is also enabled, which shown to be beneficial for faster convergence of KERNELSHAP. KERNELSHAP is written in C and is to our knowledge the fastest implementation of this algorithm.

To ensure a fair comparison, we run models on GPU wherever possible, i.e., for Neural Networks on Entacmaea and SGEMM datasets.

### F.1.3 FAST-SHAP

We tried our best to capture the full potential of FASTSHAP in predicting SHAP values in terms of speed and accuracy, to ensure a practical and grounded comparison to our method. Here are the details on training Fast-SHAP as a baseline for computing interventional SHAP values:

- **Imputer:** "Imputer" is a FastSHAP module used in the computation of neural networks' loss, acting as the value function in SHAP formula, that generates model's prediction using a strategy for treating features excluded in the subset, given the predictor and a subset of features. To compute interventional SHAP values, we use `MarginalImputer`, implemented in the original FastSHAP repo, which computes mean predictions when using the backgorund dataset's values for excluded features. Therefore, each trained FastSHAP model is specific to a fixed background dataset, as the Imputer used in its loss is. In our experiments, we use four different background datasets with multiple sizes per (real-world) dataset, which lead to training four FastSHAP models per setting.

- **Feature Subset Sampling:** We train FastSHAP models with 1, 4, and 16 feature subset samples per input. Although we did not observe monotonic improvements by increasing the number of feature subset samples, we decided to use the models trained with 16 samples per input to compare our method with, as it was mostly performing the best in terms of accuracy.

- **Paired Sampling:** We enable paired sampling in FastSHAP, which is a trick to pair each feature subset sample $s \in \{0,1\}^n$ with its complement $1 - s$, that is shown to be beneficial in reducing the variance and improving the accuracy, in both KernelSHAP and FastSHAP.

---

[3] https://github.com/slundberg/shap

- **Neural Network Architectures:** We use MLPs with three different sizes to train Fast-SHAP, and reported the results for each separately:

  - Small (2-layer): $in \times 128 \times out$.

  - Medium (4-layer): $in \times 128 \times 128 \times 128 \times out$.

  - Large (6-layer): $in \times 128 \times 128 \times 128 \times 128 \times 128 \times out$.

  We use ReLU as the activation function in all models.

- **Hyperparameters:** We use the training batch size of $64$ and FastSHAP's defaults for all other components and hyper-parameters. We train each model with early stopping and up to 200 epochs.

Table 1 shows the time required to train the FASTSHAP models used in our black-box experiments. Both FOURIERSHAP (ours) and FASTSHAP are amortized methods that have a heavier "pre-computation" step. For FASTSHAP this pre-computation appears as training an MLP that directly predicts SHAP values and for us this appears as computing a Fourier transform. When comparing FASTSHAP's initial training time to ours (reported in Figure 1), we can see our method has a considerably lower pre-computation time.

| Dataset | Black-box model | FastSHAP size | Training time (m) for background dataset size | | | | Total training time (m) |
|---|---|---|---|---|---|---|---|
| | | | **10** | **20** | **30** | **40** | |
| Entacmaea | MLP | Small | 5 | 8 | 15 | 18 | 46 |
| | | Medium | 2 | 4 | 6 | 10 | 22 |
| | | Large | 3 | 6 | 11 | 11 | 31 |
| SGEMM | MLP | Small | 47 | 130 | 249 | 173 | 599 |
| | | Medium | 55 | 52 | 162 | 128 | 397 |
| | | Large | 72 | 73 | 152 | 91 | 388 |
| GB1 | Random Forest (depth=5) | Small | 114 | 148 | 42 | 173 | 477 |
| | | Medium | 83 | 207 | 55 | 130 | 475 |
| | | Large | 54 | 102 | 155 | 184 | 495 |
| avGFP | Catboost (depth=5) | Small | 24 | 28 | 36 | 39 | 127 |
| | | Medium | 35 | 58 | 41 | 62 | 196 |
| | | Large | 11 | 20 | 22 | 25 | 78 |

Table 1: Training times of FASTSHAP models used in our black-box experiments (in minutes). Unlike FOURIERSHAP (ours), FASTSHAP needs to be separately trained for each background dataset. Part of the difference in the training times are due to the variance in the number of training epochs before the early stopping occurs. Background datasets are also of varying sizes that lead to different number of training samples per epoch.

### F.1.4 LINREGSHAP

LINREGSHAP, is a variance-reduced version of KERNELSHAP (Covert & Lee, 2020). We again use the implementation of the original writers[4]. Their implementation includes automatic detection of the convergence of stochastic sampling which is meant to speed up the algorithm by taking less samples from the black box. Furthermore, as per the default setting, we also allowed paired sampling.

To ensure a fair comparison, we run models on GPU wherever possible, i.e., for Neural Networks on Entacmaea and SGEMM datasets.

---

[4]https://github.com/iancovert/shapley-regression

### F.1.5   DEEPLIFT

For DeepLift we as well use the library of Lundberg & Lee (2017) [5] with default settings. In this setting the neural network is passed to the algorithm on a GPU to make sure this algorithm is as fast as possible.

### F.2   TREES

We fit random forests of maximum depths ranging from 3 to 8 with 20 estimators on $90\%$ of all four datasets used in the black-box setting. We compare performance of four algorithms in computing SHAP values for these random forest models; the classic TREESHAP as well as its GPU implementation [6], FASTTREESHAP [7], and our FOURIERSHAP. We always use 100 datapoints as the background data samples, and 100 datapoints to explain and compute the SHAP values for. We report the speedup of each algorithm over classic TREESHAP in Figure 3, with error bars showing the standard deviation in speedup over five independent runs.

For the first step of the FOURIERSHAP, we derive the exact sparse Fourier representation from the ensemble of trees, accessing the tree structures and using Equation 2. We also perform a pruning on the exact sparse Fourier representation derived from the ensemble of trees, and keep the frequencies with largest amplitudes that cover at least $99.95\%$ of the original Fourier spectrum's energy. We also make sure to only remove frequencies with apmlitudes smaller than $0.005$.

To run FOURIERSHAP, we use our "Precompute" implementation for Entacmaea, and "Positional" version for other datasets, given larger feature spaces and the low-degreeness of the spectrum as a result of bounded maximum depth. See Section E for implementation details. We compare the resulted SHAP values with TREESHAP results and achieve $R^2$ of at least 0.99, ensuring precise values computed by FOURIERSHAP.

### F.3   USING KERNELSHAP TO PRODUCE GROUND TRUTH SHAP VALUES

As mentioned before, in order to measure the accuracy of the SHAP values produced by our and other algorithms we need ground truth SHAP values. We use KERNELSHAP to this extent. KERNELSHAP is an approximation method for computing interventional SHAP values. This approximation becomes more precise by sampling more coalition subsets. Therefore, to generate ground truth values, for each dataset, we sample more and more and check for convergence in these values.

In order to check if KERNELSHAP values have converged to ground truth interventional SHAP values, we update the number of subset samples KERNELSHAP uses to generate SHAP values for each instance. Looking into KERNELSHAP repo [8], this is the default number of samples in the code:

```
self.nsamples = 2 * self.M + 2**11
```

Here, `self.M` is the number of indices where the instance is different from the background dataset.

To understand the convergence dynamics of KERNELSHAP, we multiply this number by multiple "sample factor"s and run the algorithm. This allows us to experiment with the number of subsets sampled, and find a sample factor that ensures convergence while avoiding unnecessary extra computation. We test sample factors in the set $\{0.02, 0.03, 0.04, 0.1, 1.0, 2.0, 10.0\}$.

For instance, for GB1 dataset, we compute the $R^2$ score of the KERNELSHAP SHAP values with different sample factors to sample factor 10 (as the ground truth). For this dataset, the default setting, i.e., sample factor $= 1$, seems good enough to make sure we are producing ground truth values and that the algorithm has converged. This is the general procedure we use for all datasets.

---

[5] https://github.com/slundberg/shap

[6] https://github.com/slundberg/shap

[7] https://github.com/linkedin/FastTreeSHAP

[8] https://github.com/shap/shap/blob/master/shap/explainers/_kernel.py

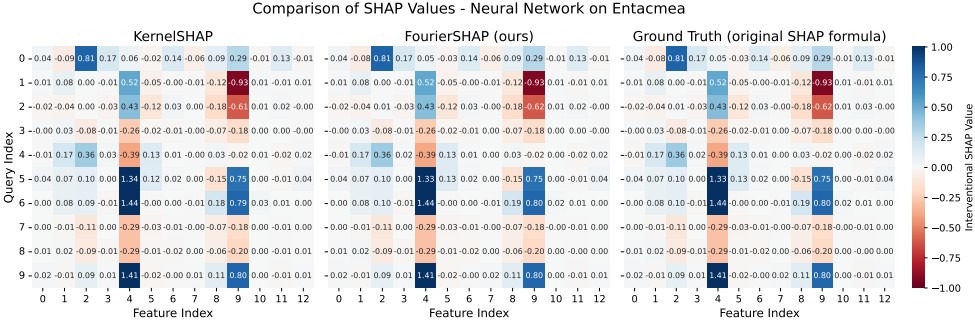

Figure 5: From left to right, 1- SHAP values generated by KERNELSHAP, 2- SHAP values generated by FourierSHAP (ours), 3- and the ground truth SHAP values computed by the original exponential SHAP formula Equation1 on the Entacmaea dataset. In 3, Computation of ground truth SHAP values using the exponential formula is possible due to the dataset containing all $2^{13}$ possible boolean feature vectors. 10 query points and a background dataset points are chosen at random and are of size 10. This figure shows that both KernelSHAP and our method compute exact ground truth SHAP values.

| Sample Factor | Runtime | $R^2$ **w.r.t Ground Truth** |
|---|---|---|
| 0.02 | 4.031 | 0.902 |
| 0.03 | 4.274 | 0.934 |
| 0.04 | 4.530 | 0.967 |
| 0.1 | 5.399 | 0.991 |
| 1.0 | 19.452 | 0.999 |
| 2.0 | 34.451 | 1.0 |
| 10.0 | 160.251 | 1.0 ($R^2$ w.r.t itself trivially=1) |

Table 2: Table of sample factor, runtime, and $R^2$ with respect to sample factor 10, on GB1 dataset.

# G  LIMITATIONS & FUTURE WORK

## G.1  EXTENDING TO CONTINUOUS FEATURES

A limitation of FOURIERSHAP is that it currently does not natively support continuous features and they have to be handled by quantization into categorical features. However, there is value in computing efficient SHAP values for models with continuous features and this is an important potential future work. In the following, we discuss how we think such an extension would work for the two classes of models we extensively covered in this work, namely trees and MLPs.

**Trees in the white-box setting.** Trees can be seen as inherently discrete structures even though they perfectly work for continuous features. By setting a threshold, i.e., a continuous number to define a split of the node, all trees are inherently in fact binary. This way of thinking gives one very simple but crude extension of the current framework: To assign to each node of the tree a binary feature specifying whether a certain continuous feature is bigger or smaller than its threshold. This would increase the feature dimension of the problem to the number of nodes in the tree in the worst case scenario. With the careful design of the transformation of continuous features into node-based binary features, this could be a potential extension for which the current work can lay a foundation.

A second and more principled and less crude way to think about the case of trees is to not use a Fourier transform that we are using now which is over $Z_2 \times Z_2 \times ... Z_2$ (n times where n is the number of features). But rather use a transform over $Z_{K_1}, \times, Z_{K_n}$ where $K_i$ is the number of distinct thresholds feature $i$ has in the tree. Coming up with a closed form solution for the SHAP values in this Fourier basis is an interesting question that we also plan to look at. It is not too far-fetched to think that we can also find a closed form solution to this discrete Fourier basis. This would perfectly handle the case of any tree. Note that by a recursive formula one can readily compute the Fourier transform of trees as well.

**MLPs.** Regarding the case of MLPs with continuous input features, one can not simply resort to the discrete Fourier transform of type $Z_k \times \ldots Z_k$ to perfectly capture this structure as it is no longer inherently discrete (like trees). This is a slightly tricky scenario because it requires us to characterize the relationship of continuous and discrete Fourier transforms of neural networks functions. In classical signal processing there are results which characterize what level of granularity in discretization is required to reconstruct continuous transform with the discrete one for a given level of error. These results depend on the Spectrum of the continuous transform. For example the case of super imposition of sinusoids, where the spectrum has a bounded domain give rise to the Nyquist rate etc.

For Neural network functions, if one had clear theoretical results on the spectral behavior of continuous transforms of neural networks it is not too hard to come up with conditions on the level of quantization required for the discrete transform to approximate them. One result that computes the spectrum appears over a multi dimensional inputs is (Theorem 1 of Rahaman et al. (2019)) but we are not aware of any spectral bias bounds on the continuous spectrum derived from this equation. However, for Neural networks with a single input dimension, bounds have been derived on the spectrum Cao et al. (2019) (in addition to the empirical results of Rahaman et al. (2019)). This intuitively means that single input Neural networks provably have a tendency to approximate functions with lower frequency sinusoids in the continuous domain. This implies that a discrete Fourier transform over $Z_K$ can approximate these function correctly for a sufficiently large discretization parameter K. This gives hope that a empirically a discrete Fourier transform of type $Z_K \times \ldots Z_K$ for a sufficiently large $K$ would also approximate neural network function with higher dimensional inputs for a large enough discretization parameter K. Therefore, an extension of our closed-form solution for SHAP to $Z_K \times \ldots Z_K$ transform could be a potential direction for the future work.

## G.2 CONDITIONAL SHAP VALUES

Another direction for future work is extending this work to compute conditional SHAP values. In this work, we only cover interventional SHAP values (the difference of interventional and conditional SHAP versions is discussed in detail in Appendix B.3).

In model-based methods like TREESHAP Lundberg et al. (2020), one can use the tree itself to approximate conditional distributions of the data in a tractable way. One idea for extending this work is to attempt to tractably find a representation of the conditional distribution of the data using a sparse Fourier approximation of the predictor. Next, computing conditional SHAP Values using this conditional approximation.

