# OpenReview forum: "Amortized SHAP values via sparse Fourier function approximation"
_ICLR.cc/2025/Conference — Submitted to ICLR 2025_

### Official Review · Reviewer_g97w · 2024-10-21

**Soundness:** 3
**Presentation:** 3
**Contribution:** 2
**Rating:** 6
**Confidence:** 3

**Summary:**

The paper proposes a fast approximation for SHAP values using sparse fourier analysis for pseudoboolean functions. It leverages the evidence that neural networks have a spectral bias that can allow the blackbox function learned by NNs to be represented by a sparse fourier approximation to derive a formula for calculating exact SHAP values that does not have a exponential sum term. For datasets considered they show orders of magnitude improvement in calculating shap values relevant to baseline methods while achieving the same accuracy (where accuracy is measured by how close the SHAP values are to values predicted by kernelSHAP).

**Strengths:**

Overall the paper is well written and covers the relevant literature well. The problem being tackled is a significant problem and speedups in calculating accurate SHAP values can be an extremely useful tool. The empirical results are convincing for the subset of problems being considered and there seems to be no problems with the theoretical underpinnings either.

**Weaknesses:**

My biggest concern with this submission is that the fact that it's only applicable to pseudoboolean functions is far too restrictive. I don't think that limitation can be overcome simply by "discretizing continuous features into quantiles" to transform them into categorical features. Post-hoc explanations like SHAP values are largely a useful area of research because they enable interpretation of real-world deployments of machine learning methods. I have trouble thinking of how many real-world problems out there can be easily tackled with boolean inputs. What are the main challenges towards extending this work to continuous valued problems? is it possible at all? at the very least a discussion is warranted. I'm afraid without that this paper, while theoretically neat would just end up without a significant impact.

In summary, the paper is lacking: a) Concrete examples of real-world problems where the boolean limitation is and isn't problematic. b) Discussion of potential challenges and ways of extending this method to continuous-valued cases. c) impact of discretization on model and SHAP performance when applied to otherwise continuous-valued datasets.

**Questions:**

My first question is described in the Weakness section and I'd love to hear more from the authors on that limiation.

Q2: One of the arguments for the assumption that NNs can be sufficiently approximated through a sparse fourier transform is presented in lines 242 to 247. From my reading of Yang and Salman it doesn't seem like they are saying that the *weak* spectral bias present in NNs is sufficient enough for the "this sample, roughly speaking, looks like a lienar combination of eigenvectors with the largest eigenvalues" line to hold true. As you point out in the footnote the spectral bias is separated out into even and odd frequencies, not higher degree and lower degree frequencies. How well are the sparse approximations in general can be relied upon to actually approximate the NNs beyond the simple cases discussed in the experiments? This also has a direct bearing on how exact the claimed "exact SHAP values" are. Consequently also, in my opinion, line 264 at best should say The aforementioned literature *suggests*, rather than "shows".

Q3: How accurate are the SHAP values? the results are relative to kernelSHAP but it might be worth setting up results on simulated data to see how well can you get at actual ground truth SHAP values.

Q4: This relates to the discretization of the data, I'm assuming there's discretization going on in some of these datasets that you included results for. How does that impact model performance? Just to get a sense of whether the underlying model itself was useful or not.

there are a few typos in the paper: please check lines 32, 33, 139, 177, 230, 231

---

> ### Author Response · Authors · 2024-11-21
>
> We appreciate your thoughtful and insightful comments. Please find our responses below.
>
>
> > My biggest concern with this submission is that the fact that it's only applicable to pseudoboolean functions is far too restrictive. I don't think that limitation can be overcome simply by "discretizing continuous features into quantiles" to transform them into categorical features. Post-hoc explanations like SHAP values are largely a useful area of research because they enable interpretation of real-world deployments of machine learning methods. I have trouble thinking of how many real-world problems out there can be easily tackled with boolean inputs. What are the main challenges towards extending this work to continuous valued problems? is it possible at all? at the very least a discussion is warranted. I'm afraid without that this paper, while theoretically neat would just end up without a significant impact.
> >
> > In summary, the paper is lacking: a) Concrete examples of real-world problems where the boolean limitation is and isn't problematic. b) Discussion of potential challenges and ways of extending this method to continuous-valued cases. c) impact of discretization on model and SHAP performance when applied to otherwise continuous-valued datasets.
>
> This is indeed a valid point that you have brought up and a point that we also care about very much. There is certainly ways that one could think of extending this framework to generalize to continuous features that we will discuss now. But before that we would like to clarify that all the datasets we used for the experiments of this paper were inherently discrete and consisted of purely categorical features (not just binary). Therefore, these were examples of real-world problems where the boolean limitation isn’t problematic.  We explain in more detail later in answer to your question 3.
>
> Back to the topic of continuous features, we already have ideas of how one can potentially generalize our results and we present them now.
>
> **Case of trees**: Our paper shows order of magnitude speedups for the case of trees. When one thinks about trees it can be seen that they are inherently discrete structures even though they perfectly work for continuous features. By setting a threshold, i.e. a continuous number to define a split of the node, all trees are inherently in fact binary. This way of thinking gives one very simple but crude extension of the current framework:  To assign to each node of the tree a binary feature specifying whether a certain continuous feature is bigger or smaller than a certain threshold (and assigning a zero or one). This would increase the feature dimension of the problem to the number of nodes in the tree in the worst case scenario. This idea does not perfectly work since some combination of zeros and ones are not allowed but we have found mechanisms of  smarter encodings that would make this transformation work. They are not mentioned in this paper and we think the current work can lay a foundation and ground of this further extension.
>
> A second and more principled and less crude way to think about the case of trees is to not use a Fourier transform that we are using now which is over $Z_2 \times Z_2 \times ... Z_2$ (n times where n is the number of features). But rather use a transform over $Z_{K_1}, \times, Z_{K_n}$ where $K_i$ is the number of distinct thresholds feature $i$ has in the tree. Coming up with a closed form solution for the SHAP values in this Fourier basis is an interesting question that we also plan to look at. It is not too far-fetched to think that we can also find a closed form solution to this discrete Fourier basis. This would perfectly handle the case of any tree. Note that by a recursive formula one can readily compute the Fourier transform of trees as well.
>
> **MLPs**: Regarding the case of MLPs with continuous input features, one can not simply resort to the discrete Fourier transform of type $Z_k \times \dots Z_k$ to perfectly capture this structure as it is no longer inherently discrete (like trees). This is slightly tricky scenario because it requires us to characterize the relationship of continuous and discrete Fourier transforms of neural networks functions. In classical signal processing there are results which characterize what level of granularity in discretization is required to reconstruct continuous transform with the discrete one for a given level of error. These results depend on the Spectrum of the continuous transform. For example the case of super imposition of sinusoids, where the spectrum has a bounded domain give rise to the Nyquist rate etc.
>
> *(Continued in the next comment)*

---

> > ### Author Response · Authors · 2024-11-21
> >
> > For neural network functions, if one had clear theoretical results on the spectral behavior of continuous transforms of neural networks it is not too hard to come up with conditions on the level of quantization required for the discrete transform to approximate them. One result that computes the spectrum appears over a multi dimensional inputs is (Theorem 1 of [1]) but we are not aware of any spectral bias bounds on the continuous spectrum derived from this equation. However for neural networks with a single input dimension bounds have been derived on the spectrum[2] (in addition to the empirical results of [1]). This intuitively means that single input Neural networks provably have a tendency to approximate functions with lower frequency sinusoids in the continuous domain. This implies that a discrete Fourier transform over $Z_K$ can approximate these function correctly for a sufficiently large discretization parameter K. This gives hope that a empirically a discrete Fourier transform of type $Z_K \times \dots Z_K$ for a sufficiently large $K$ would also approximate neural network function with higher dimensional inputs for a large enough discretization parameter K. Therefore again an extension of our closed-form solution for SHAP to $Z_K \times Z_K$ transform could potentially again handle these cases as well.
> >
> > We know these explanations do not prove that we can extend the Formula to handle the more general discrete Transform of type $Z_K \times Z_k$. However, all in all we are quite excited about the initial result that shows that spectral results of Fourier transforms can allow an computationally fast SHAP. Therefore, the result shows that an effort to find the closed form solution for a more general Fourier transform is a path worth pursuing that could lead to a potential order of magnitude speedup and something that we are enthusiastic about.
> >
> > In this paper we have primarily focused on the Boolean case because for the case of MLPs it is very easy to justify due to the extensive theoretical and experimental results presented by Yang and Salman and also Perez. However, the lack of empirical and theoretical research on MLPs with continuous input features in higher dimensions should not stop us from refining our result even further to handle the more widely applicable scenario of continuous features for the cases of Trees and Neural networks in future work. But please do not ask us to extend the current paper with these results because we think that the justification for using a high dimensional discrete Fourier transforrm for MLPs with higher dimensional inputs requires a heavy justification by itself warranting it an extensive paper in a and by itself similar to [1] and [2] covering empirical and theoretical results respectively.
> >
> > We think your idea of having at least a minimal discussion in this direction is a very good one and if the paper is accepted we can add a more concise version of the above proposed discussion to a Potential followup work section in the paper.
> >
> > >Q2: One of the arguments for the assumption that NNs can be sufficiently approximated through a sparse fourier transform is presented in lines 242 to 247. From my reading of Yang and Salman it doesn't seem like they are saying that the weak spectral bias present in NNs is sufficient enough for the "this sample, roughly speaking, looks like a lienar combination of eigenvectors with the largest eigenvalues" line to hold true. As you point out in the footnote the spectral bias is separated out into even and odd frequencies, not higher degree and lower degree frequencies. How well are the sparse approximations in general can be relied upon to actually approximate the NNs beyond the simple cases discussed in the experiments? This also has a direct bearing on how exact the claimed "exact SHAP values" are.
> >
> > The separation of frequencies to two groups of even and odd degrees is more of a byproduct of their proof. Their weak spectral bias theorem more precisely says the following:
> > "Among even frequencies, higher degree even frequencies have lower amplitudes than lower degree even frequencies. Among odd frequencies, higher degree odd frequencies have lower amplitudes than lower degree odd frequencies"
> >
> > So the main message of the result is indeed a statement on the degree of the frequency albeit in  a separated fashion among even and odds. In fact, the main focus of this paper is on the degree of the frequency as a notion of simplicity of the function the neural network presents. Their take home message is that the neural networks have a "simplicity bias". In order to formalize what "simplicity" they take the degree of their Fourier spectrum to be a notion of complexity. The weak spectral theorem, is central to their work, because it shows neural networks learn low-degree functions, and hence "simple" functions.
> >
> > We are not sure it this completely answers your question so please follow up if things are not clear.

---

> > > ### Author Response · Authors · 2024-11-21
> > >
> > > > Q3: How accurate are the SHAP values? the results are relative to kernelSHAP but it might be worth setting up results on simulated data to see how well can you get at actual ground truth SHAP values.
> > >
> > > KernelSHAP with large enough number of samples gives exact SHAP values. We understand the confusion that could arise with using KernelSHAP's SHAP values as the ground truth, since it is introduced as an approximation method. But please note that we made sure the number of subset samples we used for KernelSHAP in each experiment is large enough to guarantee KernelSHAP is generating true ground truth SHAP values and that its results have converged. We have dedicated Appendix F.3 (titled Using KernelShap to produce ground truth SHAP values) to explain this thoroughly. Specifically in lines 467-471 these points are mentioned. Moreover, we are not the first paper to be doing this. The practice of using KernelSHAP  as ground truth is seen for example in the FastSHAP paper as well [3]. We have even gone a step further in Appendix F.3 (See Table 1 and explanations) and outlined our methodology for ensuring KernelSHAP values have converged to ground truth values as explained with an example in this Appendix. In that table we show how by increasing the number of subsampled sets we are making sure the approximation algorithm has completely converged.
> > >
> > > Furthermore, even though not mentioned in the paper, we have done more thorough checks of the type you are mentioning as well. Beside that fact that we mathematically prove that our method is able to exactly compute SHAP values of a Fourier sparse function, we have empirically and thoroughly code tested this. This means we defined a Fourier sparse function with random amplitudes and Frequencies and checked that the SHAP values computed by the original exponential summation formula and our proven formula exactly match.
> > >
> > >
> > > We nevertheless want you to be assured that the formula holds. Since reviewer  trxy has also requested this we are happy to compute SHAP values using the original SHAP formula for one of our datasets, Entacmaea. This is a 13-dimensional binary dataset and therefore small enough for us to apply the SHAP original exponential summation formula (Equation (1) in our paper. Furthermore it contains all possible $2^{13}$ feature combinations in the dataset itself.
> > >
> > > We can contrast the computed SHAP values using the exponential summation formula with FourierShap's output. But before we do something like this, let us say why we expect the results to perfectly coincide. Please note for this dataset our Fourier approximation is exact ($R^2=1$) as shown in Figure 2. Our methodology which (mathematically) provably coincides with the original exponential summation Equation of SHAP, will produce the exact SHAP values (if you beleive our math is not wrong). Here are the results.
> > >
> > > > Q4: This relates to the discretization of the data, I'm assuming there's discretization going on in some of these datasets that you included results for. How does that impact model performance? Just to get a sense of whether the underlying model itself was useful or not.
> > >
> > > All four datasets we used in our experiments are discrete datasets and no further discretization is done by us to be able to fit them into our scope of work. There are many areas of ML where the data is (purely) discrete. Our SGEMM experiment is done on the data of performances of different configurations of GPUs where each parameter has a set of finite options. This is the case in many other areas where ML is used to optimise the selection of a finite set of options in different categories or hyper-parameters. Another area where the data is inherently discrete is Genomics and Bioinformatics; DNA and RNA sequences are composed of discrete nucleotides. Three other datasets we used in our experiments are of this sort. Further than areas we covered in our experiments, discrete data can be seen in other areas including survey and demographic analysis (multiple choice questions, various categorical attributes), recommender systems (ratings, user interactions, item categories), cybersecurity (event logs, attack types, protocols), robotics and control systems (discrete states, categorical actions), to name a few.
> > >
> > > As a final note, thanks for catching the typos. We  have fixed them in the text.
> > >
> > > [1] On the Spectral Bias of Neural Networks: https://arxiv.org/abs/1806.08734
> > >
> > > [2] Towards Understanding the Spectral Bias of
> > > Deep Learning: https://arxiv.org/abs/1912.01198
> > >
> > > [3] FastSHAP: https://arxiv.org/abs/2107.07436

---

> ### Comment · Reviewer_3voa · 2024-11-25
> **Acknowledgement of the authors response**
>
> We thank the author for clarifying and reorganizing the paper following the comments.
> Most of the questions and problems mentioned were addressed.
>
> The only remaining limitation in my mind is the theoretical results being on the simpler side.
> While it is possible that the writing could be more concise, the eagerness of the authors to respond to the comments made me believe that future corrections/improvement suggestions would be addressed.
>
> Due to the last point, I am more leaning toward acceptance, and while not sufficiently to bump my rating of the paper to 8 (accept) if there were more fine-tuned rating options I would have bumped my rating of the paper to 7 (weak-accept).

---

> > ### Author Response · Authors · 2024-11-26
> >
> > Dear reviewer g97w,
> >
> > We hope we have been able to address your concerns in the above response. If so we would like to ask you to please raise your score.

---

> > > ### Comment · Reviewer_g97w · 2024-11-26
> > > **Thank you for the response**
> > >
> > > I'm happy with the detailed response and will bump my score in favor of a marginally accept.

---

> ### Author Response · Authors · 2024-11-28
> **Thanking reviewer g97w for raising their score**
>
> We are happy that we could address your concerns. Thank you for reconsidering your score, it’s greatly appreciated!

---

### Official Review · Reviewer_3voa · 2024-10-27

**Soundness:** 4
**Presentation:** 3
**Contribution:** 2
**Rating:** 8
**Confidence:** 2

**Summary:**

The paper considers the problem of computing the SHAP values of machine learning models, which is essentially the significance of each feature.

This problem is well-motivated, has a significant amount of work, and even has a legal interest.

The direction is rather interesting and seems novel as well.

**Strengths:**

Extensive experimental results, that include various models.

The problem is very well-motivated, as stated in the introduction Section.

The paper is fully contained since I have understood the main parts of the paper without significant prior knowledge on the subject.

The direction seems rather novel and interesting.
That is, provide an exact bound on the SHAP value of *Fourier representation*, which can approximate (or even give exact representation) many models, and in turn yields an approximation to the SHAP value or an exact value.

**Weaknesses:**

The theoretical results are on the simpler side, specifically the proofs span around 2.5 pages and seem to be on the simpler side.
While there is absolutely no need to overcomplicate the proofs and short proofs have merit, I believe that it should be noted.

The writing seems to require some improvement.
Specifically, I felt that the abstract, introduction, and background were on the cumbersome side and should be shortened.
As a consequence, the introduction only "moved me over the fence" on the significance of the problem and did not truly excite me.
It also seemed to me that the abstract is rather long and is somewhat repeated in the introduction.

Additional writing suggestions are included in the questions section.

**Questions:**

The paper answered the suggested problem and I have no significant questions.
Nonetheless, regarding the approximation of neural networks, I am intrigued by the exact values (as a function of the approximation) of the representation for common networks (say AlexNet).
It seems that this is an active research area in itself, thus perhaps there are yet exact values for such networks.
If there are such results consider including them since it could give a good picture of the complexity of approximating (with theoretical guarantees) the SHAP value via the proposed method.

Regarding the writing, I suggest the following changes:

* Consider rewriting the abstract, introduction, and background to be more concise since I found it to be on the cumbersome side.

* When mentioning the "exponential sum" in the second paragraph of page two consider giving a reference to the full expression on page 3.
I am unsure if this is due to me being unfamiliar with the matter, but noting the sum over the exponential number of combinations simply as "exponential sum" rather confused me initially.

* The first line of Section 2.2 seems to require rewriting.

* I would prefer the machine properties in the experimental section, and for you to include more details such as CPU and RAM.

## In the proof of Lemma 1:

* Consider Changing "One can see by checking for different values" to something along the lines of "By checking all the possible 8 values" since it is needed to check for all the values, and the current writing seems to unnecessarily suggest corner cutting to the reader.

* I believe that both Equations i and ii should be proven before use in Equation 8.
This would make the proof more organized.

* Fully prove Equation i and not leave it to the reader, especially since the cancelations can be demonstrated rather effectively.

* While Equation ii is correct, and I liked that it was a combinatorial proof, the writing is rather confusing, consider rewriting it.

---

> ### Author Response · Authors · 2024-11-21
>
> Thank you for your thorough review and your great attention to the details. It is delightful to see you found the text fully contained and highlighted the originality of the direction of our workWe are happy to discuss the weaknesses you mentioned and go through the questions.
>
> > The theoretical results are on the simpler side, specifically the proofs span around 2.5 pages and seem to be on the simpler side. While there is absolutely no need to overcomplicate the proofs and short proofs have merit, I believe that it should be noted.
>
> We agree and also believe there is no need to complicate proofs for the sake of lengthening them.
>
> > The writing seems to require some improvement. Specifically, I felt that the abstract, introduction, and background were on the cumbersome side and should be shortened. As a consequence, the introduction only "moved me over the fence" on the significance of the problem and did not truly excite me. It also seemed to me that the abstract is rather long and is somewhat repeated in the introduction.
>
> We understand your concern. Since the paper requires quite some background explanation especially on the Fourier approximations, which are not typical in papers in this field, the abstract and introduction has become longer than the typical paper. If accepted we will go over these sections again and try to decrease redundancies. We are very much open to suggestions from you about parts that you felt like were unnecessary and/or pedantic/boring.
>
> > Nonetheless, regarding the approximation of neural networks, I am intrigued by the exact values (as a function of the approximation) of the representation for common networks (say AlexNet). It seems that this is an active research area in itself, thus perhaps there are yet exact values for such networks. If there are such results consider including them since it could give a good picture of the complexity of approximating (with theoretical guarantees) the SHAP value via the proposed method.
>
> We thank the reviewer for this suggestion and we think that indeed that this is indeed an interesting and impactful direction to take. One can think of CNNs, as roughly speaking, MLPs that slide along the data channels spatial positions. Therefore, one could apply the same methodology and spectral bias results that we presented here. However the extension requires more careful thinking and a more work. We are not aware of any results that try to carefully extend Spectral bias results (in the spirit mentioned above) to the case of CNNs. Perhaps the reason could be that, there was not much practical value to do so. We believe our paper provides at least some incentive, that these spectral bias results can be concretely used for some important down stream task, i.e. estimating SHAP values.
>
> Regarding your question about other results in the literature for how well deep shap methods approximate the true SHAP values. Again to the best of our knowledge the goto method for this is  DeepLift and FastSHAP. Which we have included as a baseline. We have also documented the $R^2$ scores we get for the faithfulness of their SHAP values to ground truth values. As mentioned in the paper, unlike FourierSHAP, for these methods one can not reliably have a fine-grained control over the level of approximation and trade it off with computation. In Fourier SHAP there is at least a proxy for the level of accuracy of the SHAP values namely the Fourier approximation error. Moreover DeepLift is not model agnostic and assumed full access to the model architecture.
>
> > When mentioning the "exponential sum" in the second paragraph of page two consider giving a reference to the full expression on page 3. I am unsure if this is due to me being unfamiliar with the matter, but noting the sum over the exponential number of combinations simply as "exponential sum" rather confused me initially.
>
> That is indeed a valid point, and the whole line can be written clearer. We updated the first line of that paragraph and it is reflected in the updated pdf.
>
> > The first line of Section 2.2 seems to require rewriting.
>
> Thanks for the catch. We rewrote the first sentences as follows:
> *Here we review the notions of the Fourier basis and what we mean by sparsity. We will later use sparse Fourier representation of functions to compute SHAP values. In this work, ...*
>
> Hope it is more clear now.
> > I would prefer the machine properties in the experimental section, and for you to include more details such as CPU and RAM.
>
> All experiments are ran on servers with Intel(R) Xeon(R) CPU E3-1284L v4 @ 2.90GHz, restricting the memory/RAM to 20 GB, which was managed with Slurm. We will add this detail to Appendix F, Experiment Details. We will make sure to add more details about the setup in the main text as well.

---

> > ### Author Response · Authors · 2024-11-21
> >
> > > Consider Changing "One can see by checking for different values" to something along the lines of "By checking all the possible 8 values" since it is needed to check for all the values, and the current writing seems to unnecessarily suggest corner cutting to the reader.
> >
> > Thanks for pointing this out. We did not intend to corner cut the proof but rather were trying to not exhaust the minor details. However, we like your suggestion and rewrite that line more precisely:
> >
> > "By checking all 8 possible combinations of $x_i, x_i^*, f_i \in \{0,1\}$, one can see that $(-1)^{f_ix^*_i} - (-1)^{f_ix_i} = 2f_i(x_i-x_i^*)$. This is simply because the two exponents only differ when $f_i=1$ and $x_i\neq x_i^*$."
> >
> > >* I believe that both Equations i and ii should be proven before use in Equation 8. This would make the proof more organized.
> >
> > > * Fully prove Equation i and not leave it to the reader, especially since the cancelations can be demonstrated rather effectively.
> >
> > > * While Equation ii is correct, and I liked that it was a combinatorial proof, the writing is rather confusing, consider rewriting it.
> >
> > Regarding your comments on the proof of Equations i and ii. This led us to rewrite the two paragraphs discussing these two. We thank you for taking the time to thoroughly review our proofs and hope the new version is clearer. We included the full proof for Equation i instead of leaving it to the reader, and tried to make the proof for Equation ii clearer. Please make sure to check it out in the revised version of the pdf.
> >
> > Thank you for your invaluable and detailed feedback. We believe our revisions based on your suggestions have improved the proofs, making them clearer and easier to follow. Please feel free to share any additional specific suggestions that you think would be helpful for us to address.

---

> > > ### Author Response · Authors · 2024-11-26
> > >
> > > > We thank the author for clarifying and reorganizing the paper following the comments. Most of the questions and problems mentioned were addressed.
> > > >
> > > >The only remaining limitation in my mind is the theoretical results being on the simpler side. While it is possible that the writing could be more concise, the eagerness of the authors to respond to the comments made me believe that future corrections/improvement suggestions would be addressed.
> > > >
> > > >
> > > > Due to the last point, I am more leaning toward acceptance, and while not sufficiently to bump my rating of the paper to 8 (accept) if there were more fine-tuned rating options I would have bumped my rating of the paper to 7 (weak-accept).
> > >
> > > We are glad that we could address your concerns about our work.
> > >
> > > We would greatly appreciate any additional suggestions or concerns you believe could further strengthen the work to meet the standard of a clear accept.
> > >
> > > Given that the option for a score of 7 is not available this year, we hope to have clarified why we believe the work aligns more closely with an 8 than a 6 :D

---

> > > > ### Author Response · Authors · 2024-11-29
> > > > **Thanking reviewer 3voa for increasing their score**
> > > >
> > > > Thank you for reevaluating and increasing your score — we just noticed! We greatly appreciate your thoughtful review and constructive suggestions, which were invaluable in improving our manuscript.

---

### Official Review · Reviewer_GoTf · 2024-10-29

**Soundness:** 3
**Presentation:** 3
**Contribution:** 2
**Rating:** 5
**Confidence:** 4

**Summary:**

The paper presents a novel method, FourierSHAP, that amortizes the computations of SHAP values. FourierSHAP can be employed to explain tree-based models or black-box neural network models. The proposed method is composed of two phases. The first phase, which is the most computationally expensive, extracts a compact Fourier representation approximation using a sparse Fourier approximation
algorithm. In the second phase, the SHAP values are computed using a closed-form expression that avoids the exponential summation operation over all possible coalitions of features. The proposed method showed an order of magnitude improvement in running time over the baseline method, KernelSHAP.

**Strengths:**

1- The approach presented in the paper to compute the SHAP values is novel and theoretically grounded.

2- The paper is mostly clear and understandable, although the description of Fourier representations can be further clarified.

3- The discussed problem, i.e., reducing the computational cost of SHAP values to explain black-box models, is important.

4- The presented solution provides a new perspective to compute the SHAP values, which can benefit the explainable machine learning community.

**Weaknesses:**

1- The proposed method is limited to binary features or categorical data that can be presented using one-hot encoding.

2- FourierSHAP is not a model-agnostic explanation method, e.g., KernelSHAP, i.e., does not explain any black-box model but is limited to either tree-based models or neural networks (or any sparse and low-degree functions).

3- The experiments are limited, including only four datasets.

4- The metric used to evaluate the accuracy of the SHAP approximations does not adequately capture how closely the approximated values align with the ground truth, nor does it assess the agreement in feature ranking based on their relative importance.

**Questions:**

1- Line 105 says, ``This was not formerly possible with these two black box methods."
which two black-box methods is the line referring to?

2- In the experiments, why were not all the algorithms trained and explained on all the datasets for evaluation?

3- In the evaluation of the SHAP value accuracy, why just 4 query points were selected to be explained?

4- As I understand it, all the evaluation datasets are for regression problems. Is FourierSHAP applicable to classification problems as well?

---

> ### Author Response · Authors · 2024-11-21
>
> Thank you very much for your thorough comment and the subtle questions you asked. We are delighted that you find our work important and beneficial for explainable ML community. We are happy to address some of the points mentioned and answer your questions.
>
> >1- The proposed method is limited to binary features or categorical data that can be presented using one-hot encoding.
>
> We have addressed this concern quite extensively in our answer to reviewer g97w Q1. We ask you to kindly read this section and tell us what you think. There we mentioned our ideas for extending this work to continuous domain, as well as highlighting the scope of binary / categorical data.
>
> > 2- FourierSHAP is not a model-agnostic explanation method, e.g., KernelSHAP, i.e., does not explain any black-box model but is limited to either tree-based models or neural networks (or any sparse and low-degree functions).
>
>
> We believe there has been a misunderstanding regarding the model-agnostic nature of FourierSHAP. The only requirement for applying FourierSHAP is query access to the model. There are no assumptions that we know anything more about the model's internal structure (such as structure of the MLP or the tree), and the method can be used with any black-box model as long as you can give input x and query the value i.e. get $f(x)$
>
> The reason we specifically mentioned tree-based models and MLPs in our paper was to highlight the expressive class of functions namely the Fourier sparse functions. Both of these models are representative examples of functions that have sparse Fourier representations. This was not intended to imply that FourierSHAP is limited to these models. In fact, FourierSHAP can be applied to any black-box model, regardless of its type or structure.
>
> For example, in our black-box experiments with GB1 and avGFP, we applied FourierSHAP without relying on any structural knowledge of the MLPs. For these MLPs, we make queries to the model without accessing its weights or architecture. Similarly, while we utilized the tree structure for extracting the sparse Fourier transform from tree-based models, we could have treated the tree as a black box and still applied the Fourier methodology. This would have taken longer, but it would have worked. These experiments show that FourierSHAP is truly model-agnostic.
>
> It is also important to note that many model-agnostic explanation methods, such as LIME, inherently assume some form of regularity in the model to make the method effective. For instance, LIME assumes that the model can be locally approximated by a linear function. Similarly, FourierSHAP relies on the assumption that a model can be well-approximated by a sparse Fourier representation.
>
> > 3- The experiments are limited, including only four datasets.
>
> We appreciate the reviewer’s concern regarding the number of datasets used in our experiments. While we only included four datasets in the current study, these datasets were carefully selected to showcase the versatility and effectiveness of FourierSHAP across different types of models (trees and MLPs) and domains (biology and discrete GPU hyperparamer performance). We believe that this initial set provides evidence of the method's applicability.
>
> > 4- The metric used to evaluate the accuracy of the SHAP approximations does not adequately capture how closely the approximated values align with the ground truth, nor does it assess the agreement in feature ranking based on their relative importance.
>
>  Given the fact that on our 4 datasets are all linear regression datasets we think that the magnitude of the SHAP importance value is also of importance (in addition to the rank). Therefore the metric should capture that as well. If the datasets were classification, perhaps the ranking would have been a better metric as there the magnitude is less meaningful.
>
> > 1- Line 105 says, ``This was not formerly possible with these two black box methods." which two black-box methods is the line referring to?
>
> DeepLift and FastSHAP mentioned in line 102. We will try to rewrite line 105 to avoid the confusion.
>
> > 2- In the experiments, why were not all the algorithms trained and explained on all the datasets for evaluation?
>
> There was no specific reason here other than us not wanting to run too many experiments. We randomly assigned one prediction model (tree, MLP etc. ) to each dataset. We could have done what you said as well. Not sure if we could fit 12 sets of result (as opposed to the current 4) in the main body of the paper though and most would go to the appendix

---

> > ### Author Response · Authors · 2024-11-21
> >
> > > 3- In the evaluation of the SHAP value accuracy, why just 4 query points were selected to be explained?
> >
> > Quoting from lines 428-430 of the script; ``For each model to be explained, we choose four different values for the number of background samples and four different values for the **number of query points** to be explained, resulting in a total of 16 runs of FourierSHAP for each model.''
> >
> > So the number 4 does not simply refer to the number of query points. We  used background and query datasets of 4 different sizes, namely, (10, 20, 30, 40), for both background and query sets. This resulted in 16 different runs. The randomness in the results, over these 16 runs, provide error bars in Figure 2.
> >
> > > 4- As I understand it, all the evaluation datasets are for regression problems. Is FourierSHAP applicable to classification problems as well?
> >
> > The sparsity result on the tree Fourier transform holds for any tree whether it was trained for a regression or classification task. The same can be said for MLPs. The spectral bias results are derived using NTK theory. Neural Tangent Kernel (NTK) is not limited to just the Mean Squared Error (MSE) loss function. While NTK has been most commonly analyzed in the context of regression problems with MSE loss, it is a general framework that can be applied to a variety of loss functions in different machine learning tasks, including classification problems (with cross-entorpy loss). But we have also focused on the regression task at hand.

---

> ### Comment · Reviewer_GoTf · 2024-11-23
>
> Thank you for providing clarifications and answers to the raised questions. After going through the authors' answers, I remain concerned that the empirical investigation is very limited and does not provide sufficient evidence for the claims on the performance of FourierSHAP. For example, in response to Weakness #1 (that FourierSHAP is limited to binary features or categorical data), the authors proposed potential ways to generalize the approach to continuous features. However, such "ways" should be evaluated first and listed in the experiments to be considered as possible solutions. Additionally, the authors clarified that FourierSHAP is model-agnostic and can be applied to any black-box model, but this was not obvious in the paper. For instance, I could not find experiments involving black-box models other than tree-based models and feedforward neural networks, which raises more questions on the quality and time complexity of FourierSHAP on models beyond tree-based models and feedforward neural networks. **I tend to think the paper should be rejected until it provides a more extensive empirical investigation.**

---

> ### Author Response · Authors · 2024-11-28
> **Response to Reviewer GoTf (1/3)**
>
> Thank your for taking the time to read our response. We are disappointed that our rebuttal has caused more confusion. Please allow us to make another attempt at addressing your concerns to hopefully clear up any points that we mis-communicated. You have raised a lot of points so let us dissect your response into different parts. There is quite a lot to unpack here so let's go step by step.
>
> > For example, in response to Weakness #1 (that FourierSHAP is limited to binary features or categorical data), the authors proposed potential ways to generalize the approach to continuous features. However, such "ways" should be evaluated first and listed in the experiments to be considered as possible solutions
>
> First of all, let us discuss "ways". Regarding "ways", we believe you are referring to our response to reviewer "g97w", who has by the way raised their score after reading the response. As you and the other reviewer have kindly pointed out, our algorithm does not natively support continuous features. However we have made no such claims anywhere in the paper that we are addressing the continuous case and we, do not want to deny that the work at its current state has this limitation. At the proposal of reviewer "g97w" and "trxy" we have now added a limitations sections emphasizing the fact that we do not support continuous features (Appendix G). Nevertheless, as you and reviewer g79w were quite rightfully concerned about extension paper to continuous we have written a **potential** future direction in response to this reviewer. And given the positive response of reviewer "g97w" we have now added this as a future direction section in Appendix G as well. That being said, if you read our response there, these potential future directions require extensive justification involving theoretical and empirical on the spectral biases of neural network with continuous inputs that would pave the way for us to also extend this algorithm. This would be the subject of not just one but a few research papers and is not trivial, but as we have mentioned, as potential future work, it is not too far-fetched either.
>
> Now the remaining concern here is that without handling continuous features, is the contribution of our paper worthy. And we will address this after discussing two of you other concerns first.
>
> > Additionally, the authors clarified that FourierSHAP is model-agnostic and can be applied to any black-box model, but this was not obvious in the paper
>
> We politely disagree here. There are 2 sub-cases:
> A- If you are saying that we have not been clear in the presentation of the paper that we are claiming to present a model-agnostic (a.k.a black box) algorithm. These claims appear as early as line 3 in the abstract:
> Quote: "We cover both the model-agnostic (black-box) setting, where one only has query access to the model.. "
> Other places include but are not limited to lines 72, 93, 94, or the whole experiment section titled black-box staring on line 421.
>
> B- If you are saying that it is not obvious that our algorithm is a black-box method we also politely disagree. Let us explain more clearly: The only requirement for applying FourierSHAP is query access to the model i.e. that you can give an input x and see the value of the predictor f(x). The fact that FourierShap is a black-box does not even require us to present experiments. You can see that by understanding the algorithm itself. The first stage of our algorithm, which is extracting a Fourier transform only assumes query access (as clarified numerous times see for example line 72).
> So unless the reviewer somehow thinks we cheated in our experiments and did something that we did not claim to do in the paper, there should be no doubt we are only querying the black box. If accepted the code release would also help alleviate the reviewers concerns.

---

> ### Author Response · Authors · 2024-11-28
> **Response to Reviewer GoTf (2/3)**
>
> Going on to the next concern:
>
> > For instance, I could not find experiments involving black-box models other than tree-based models and feed forward neural networks, which raises more questions on the quality and time complexity of FourierSHAP on models beyond tree-based models and feed forward neural networks
>
> After reading this sentence and connecting it with the last quote, we think that the reviewer, when saying our algorithm is not model-agnostic, means that it can not be applied to black boxes other than trees and neural networks.
> We would like to politely disagree here too. In order to get our point across we need some background. First of all let us consider the most widely used and well-known model-agnostic algorithm kernel shap with over 10000 citations. At the heart of of this model-agnostic method, there is an approximation and this approximation can come with guarantees. In the case of kernel shap this appears as asymptotic statistical guarantees. Basically for a large enough number of sampled subsets kernel shap recovers the ground truth shap values (by ground truth shap values we mean ones that can be computed by the exponential summation equation 4.1). In the worst case scenario though,  when we have $n$ features, one has to sample all possible $2^n$ subsets. It is the same for our case: one can get better and better approximations of the black box by increasing the sparsity parameter. In the worst case one can get a perfect approximation using $2^n$ queries to the black box i.e. computing the whole Fourier spectrum.
>
> But there is hope. The authors of kernel shap, proved that for simple models such as linear predictors in the form of $f(x_1, \dots, x_n)=a_1 x_1+ \dots + a_n x_n$, kernel shap exactly recovers ground truth shap values without taking exponentially many samples $n$. Please see section 4.2 of [1] for a list of specific models kernel shap is guaranteed to produce ground truth values for.
> This might sounds like a very weak guarantee, but please note without regularity constraints on the prediction function $f$ one can not hope to overcome the exponential summation of Equation 4.1.
>
> Now what we have done for our model-agnostic algorithm is shown that for a class of significantly more complex predictor models than linear models, namely trees and MLPs our algorithm is able to compute the ground-truth shap values  (exponential summation Eq4.1) precisely. This is possible due to the fact that both these predictors show regularity and structure when you take their Fourier transform, namely they are sparse. We have shown how to exploit this regularity to overcome the exponential summation of shap value computation. We would like the reviewer to contemplate the fact that this is the main contribution of our work. Even though it only applies to discrete features, and has not yet been extended to continuous features, it provides a big leap in the realm of model-agnostic algorithms. Note that trees capture linear predictors as a very special case (Any linear predictor over $n$ variables is a summation of $n$, depth=1 trees)
>
> Now lets go to our experiments. Since our algorithm is more suited to discrete data, we have chosen datasets that only contain categorical features. We have trained neural networks and trees on these datasets. If the reviewer has an ML predictor model that they think works well on these datasets we are happy to oblige, but we think that for these types of datasets trees and MLPs are probably the two most widely used models.
>
> Our choice of experiments are also justified because they show that our claims about guarantees to produce ground truth SHAP values are true.

---

> > ### Author Response · Authors · 2024-11-28
> > **Response to Reviewer GoTf (3/3)**
> >
> > Now the only concern the reviewer might have left is that say we have a model that is not a neural net or a tree. How are you sure that your model-agnostic algorithm is better than others, as you have not provided enough extensive empirical investigation.
> >
> > I- The reviewer could ask the same from the authors of kernel shap: that you have not provided empirical evidence that your model works beyond linear predictors in a computationally efficient manner. Which is a perfectly valid question: as the author can see our algorithm, FourierShap. has a few order of magnitude speedup over kernel shap for the case of a non-linear predictor like an MLP, because of its smart design.
> >
> > The author's of kernel shap would reply that yes we don't make any claims there but the method can nevertheless be applied and is guaranteed to work for large enough number of samples subsets. Our algorithm is the same, it is no longer guaranteed to work in those cases but would work perfectly fine for a large enough sparsity parameter $k$.
> >
> > In fact, one could speculate that since our method is theoretically and empiricially (as shown by our experiments) more computationally efficient on a much larger class of models (namely trees and MLPs) when compared to KernelShap, it makes sense that it might also be faster on that black box.
> >
> > II- There is also a slightly deeper more philosophical answer to this question, that we will present but please do not quote us saying we should back it by extensive empirical evidence. Perhaps the reason that, there are a fixed class of models namely trees and neural networks that are so widely used on tabular data is that both of these models have a tendency to be sparse in the Fourier domain so that they do not overfit. This is assuming the notion of complexity (overfitting) of a function to de directly related to its sparsity in the Fourier domain [1, 2]. So basically what we are saying is that, any useful black box that is not over-fitted and is not too "complex" in the Fourier sense would work well under this algorithm.
> >
> > Now finally as promised going back to the continuous feature question. Yes we lack natively the support of continuous features. But we would like the reviewer to appreciate the deep connection that has been made in the discrete setting between two completely irrelevant fields: Spectral theory of neural networks and Efficient computation of SHAP values. Although it is not obvious what continuous spectrums of neural networks look like it is very much an active field and makes us excited about the future. Finally we would like to mention that not all datasets are continuous. For example all the datasets we provided in the expeirments section consisted solely of categorical features. Our SGEMM experiment is done on the data of performances of different configurations of GPUs where each parameter has a set of finite options. This is the case in many other areas where ML is used to optimize the selection of a finite set of options in different categories or hyper-parameters. Another area where the data is inherently discrete is Genomics and Bioinformatics; DNA and RNA sequences are composed of discrete nucleotides. Three other datasets we used in our experiments are of this sort. Beyond areas we covered in our experiments, discrete data can be seen in other areas including survey and demographic analysis (multiple choice questions, various categorical attributes), recommender systems (ratings, user interactions, item categories), cybersecurity (event logs, attack types, protocols), robotics and control systems (discrete states, categorical actions), to name a few.
> >
> > [1] A Fine-Grained Spectral Perspective on Neural Networks: https://arxiv.org/abs/1907.10599
> >
> > [2] Deep learning generalizes because the parameter-function map is biased towards simple functions: https://arxiv.org/abs/1805.08522

---

> > > ### Comment · Reviewer_GoTf · 2024-11-29
> > >
> > > Thank you for the detailed response. I want to clarify that the burden of proof cannot be handed over to the reviewer. Therefore, it is not my duty to come up with/suggest models that work well on the mentioned datasets nor to ask questions to the authors of previous work.
> > >
> > > I appreciate the novelty of the work, and I believe that it provides a new perspective to compute the Shapley values, which benefits the explainable machine learning community.
> > >
> > > However, my main concern is the limited experimental setup. There are many publicly available categorical data that can be added to the experiments, e.g., the Pokerhand dataset and the Dota2 Games Results dataset (available at https://www.openml.org/). Therefore, I do not have a strong opinion against the method being limited to binary representations, and I do not consider it sufficient grounds to reject the paper.
> > >
> > > The main reason for me to reduce the score is that the experimental setup is very limited in terms of the number of datasets and the evaluated models.

---

> ### Author Response · Authors · 2024-12-01
>
> Thanks for the response. We are happy that you find our work novel and beneficial for the XAI community. We are happy to have addressed most of the concerns you have raised during the rebuttal phase. We can see that your only main remaining concern is that we have not evaluated our model on enough datasets. The datasets that you mentioned in the gaming area are definitely interesting ones for the application of FourierSHAP. We did not intentionally prioritize experiments/datasets of one area over another.
>
> Before getting into datasets, we would like to emphasize the fact that we have presented a completely new and novel way of computing SHAP values that did not exist before. Therefore a lot of the writing of the paper has gone to presenting the soundness of this method and justifying why it makes sense with guarantees. And significant work in the experiments is in lines with those explanations and just solidifying that this new algorithm is doing what it claims to be doing. The fact that no reviewer is questioning whether the algorithm is correct shows that the experiment section is doing its job well in our humble opinion. We were only asked by reviewers to run an experiment to show we are recovering ground truth SHAP values, which we happily provided for them. That being said, not only have we successfully shown that the new algorithm is sound and correct we have showcased its performance on 4 different datasets, with order of magnitude improvements in runtime compared to other methods.
>
> We very much appreciate, agree, and sympathize with the reviewer's concern for empirical evaluation. We think the number of datasets we have provided is reasonable though, when compared to other SHAP methods presented in top ML conferences. In order to alleviate the reviewers concern that this is a standard number of datasets to experiment on and that this is in line with other similar works proposing a SHAP value computation methods, we have conducted a list of other papers that are similar to ours and published in top ML conferences.
>
> | **Method**                                      | **Venue**                        | **Experimented Datasets**                                                                   |
> |-------------------------------------------------|----------------------------------|---------------------------------------------------------------------------------------------|
> | RKHS-SHAP [1]                                   | NeurIPS 2022                     | Diabetes                                                                                    |
> | GP-SHAP [2]                                     | NeurIPS 2022 (Spotlight)         | California, Breast cancer, Diabetes                                                         |
> | Linear TreeSHAP [3]                             | NeurIPS 2022                     | Adult, Conductor                                                                            |
> | Interventional TreeSHAP [4]                     | AAAI 2023                        | 7 Classic Tabular Datasets: pu_act, isolet, Ailerons, house_16H, sulfur, superconduct, year |
> | TreeSHAP [5]                                    | Nature Machine Intelligence 2020 | Three medical datasets: NHANES I mortality, Chronic Kidney Disease, Hospital Duration       |
> | Original TreeSHAP, KernelSHAP, and DeepLIFT [6] | NIPS 2017                        | An original user study, MNIST                                                               |
> | FastSHAP [7]                                    | ICLR 2022                        | News, Census, Bankruptcy, CIFAR10, Imagenette                                               |
> | Linear Regression SHAP [8]                      | AISTATS 2021                     | Census, Bank Marketing, German Credit, Breast cancer                                        |
>
> We hope we are able to portray to the reviewer that the current set of experiments is justified. We would like to ask the reviewer to again consider the work as a complete package of a proposed, theoretically-backed method with clarified domains of application and guarantees that is backed up by a set of experiments that convey the proposed method in a concise but sound manner.
>
>
>
> ## References
>
> [1] RKHS-SHAP: https://openreview.net/pdf?id=gnc2VJHXmsG
>
> [2] GP-SHAP: https://openreview.net/forum?id=LAGxc2ybuH
>
> [3] Linear TreeSHAP: https://openreview.net/forum?id=OzbkiUo24g
>
> [4] Interventional TreeSHAP: https://ojs.aaai.org/index.php/AAAI/article/view/26322
>
> [5] TreeSHAP: https://www.nature.com/articles/s42256-019-0138-9
>
> [6] A Unified Approach to Interpreting Model Predictions: https://papers.nips.cc/paper_files/paper/2017/hash/8a20a8621978632d76c43dfd28b67767-Abstract.html
>
> [7] FastSHAP: https://openreview.net/forum?id=Zq2G_VTV53T
>
> [8] Linear Regression SHAP: https://proceedings.mlr.press/v130/covert21a.html

---

### Official Review · Reviewer_trxy · 2024-10-30

**Soundness:** 2
**Presentation:** 1
**Contribution:** 2
**Rating:** 3
**Confidence:** 4

**Summary:**

The paper proposes FourierSHAP as a novel way to amortize Shapley value (SV) computation for XAI use cases. Therein, the work sits between traditional approximation methods such as KernelSHAP or Permutation Sampling and model-based amortized methods such as FastSHAP. In a way, FourierSHAP can be compared to model-specific methods like TreeSHAP or RKHS-SHAP. FourierSHAP transforms the black-box model to be explained into a fourier representation. Then, based on interesting and novel theoretical results, FourierSHAP computes SVs for the transformed model efficient.

## References for this Review
- [1] RKHS-SHAP: https://openreview.net/pdf?id=gnc2VJHXmsG
- [2] GP-SHAP: https://openreview.net/forum?id=LAGxc2ybuH
- [3] Linear TreeSHAP: https://openreview.net/forum?id=OzbkiUo24g
- [4] Interventional TreeSHAP: https://ojs.aaai.org/index.php/AAAI/article/view/26322
- [5] TreeSHAP: https://www.nature.com/articles/s42256-019-0138-9

**Strengths:**

- **Novel Theoretical Results:** Theorem 2 and Lemma 1 are (to the best of my knowledge, though I did not check [1,2] in detail) novel and very interesting results. This result can be useful for different kernel-based model classes and like the authors demonstrate even to create high-fidelity surrogate representations of black box function. This is a good contribution.

- **Potentially Good Surrogate**: From Figure 1 and the theory of the paper, it seems like the Fourier representation of the tree models seems to be a good surrogate model for approximating tree structures (high $R^2$ in Figure 1). I think this is a nice experiment and insight whcih begs the question if this holds for even more complicated black box models and functions.

**Weaknesses:**

While the paper proposes interesting theoretical results strengthening the body of research concerned with efficient computation of Shapley value based explanations, it ultimately falls short to present a convincing picture of their method FourierSHAP. This problem arises from the fact that the work evaluates FourierSHAP in a very limited way missing key baseline comparisons, which would help researches better understand the strengths and weaknesses of FourierSHAP. The method itself has inherent restrictions that are not well explored or discussed:

- **Restrictive Application Settings:** _First_, FourierSHAP can only applied when input spaces are binary. This means numerical features cannot be represented. This is a big limitation of the method, which obvious competitors like [1-4] do not have, which reduces the applicability of FourierSHAP in settings with numerical data. I am sure workarounds and fixes exist (binning and binarization), however this is left unexplored by the paper. _Second_, FourierSHAP seems to work only with very shallow neural networks (i.e. 3 layers with 300 neurons), which definitely is not realistic given modern architecture sizes. Usually, for XAI use cases testing with smaller networks is ok for model-agnostic methods, as the focus lies in showing that such methods can work with all sorts of models. However, given that FourierSHAP relies on finding a sparse Fourier representation of the underlying model, the aforementioned argument is invalid. This Fourier representation will most definitely be drastically different from smaller sized networks. This needs to be theoretically or at least empirically explored. Otherwise, I am doubtful the results generalize to large scale networks.

- **Lacking Empirical Evaluation**: While FourierSHAP can be seen as a model-agnostic method for computing SVs, the experimental section and focus of the paper lies with **tree ensembles** and **very small neural networks**. Also strong focus of the paper is to show how SV computation can be amortized, i.e. made more efficient. Therein, the paper **needs to compare itself with better baselines**. These baselines include _TreeSHAP_ [5] which was made even more efficient with _Linear TreeSHAP_ [3] and extended to also account for _interventional SVs_ in [4]. As the core concept of the method is a kernel representation of the underlying model to be explained, a comparison to kernel based SV computation methods is important as well. This would include works like _GP-SHAP_ [2] or _RKHS-SHAP_ [1]. I really don't think DeepLIFT or potentially even FastSHAP is the right baseline to compare against here. Lastly, computing ground-truth SVs with an approximation based method like KernelSHAP is an okay proxy, but in smaller feature settings like the _Entacmea_ (13 features) dataset used in this paper or potentially synthetic settings, real ground truth values can be obtained through brute force or clever tricks with knowledge about the underlying data generating process / black box function.

- **Missing Limitations Section**: The paper does not contain a limitation section discussing FourierSHAP's drawbacks and avenues for interesting future work.

- **Poor Related Work**: While the paper does include a related work section in the appendix, key works [1-4] are missing and not delineated from. Given the above mentioned critique with the evaluation, this is problematic.

- **Presentation can be Refined**: The papers could be presented better. The work is more verbose than it needs to be. This starts in the abstract which is overly detailed and too long. The paper looses itself in discussing aspects that are in my opinion secondary and can be postponed to the appendix, like the parallelization and GPU acceleration. The paper contains multiple typos.

**Questions:**

- How does the Figure 1 look if the underlying model is a large neural network that potentially operates on text (text is also binary)?
- How is FourierSHAP different from [1] and [2]?

---

> ### Author Response · Authors · 2024-11-21
>
> We would first like to thank you for your comprehensive, thoughtful and detailed feedback on the work. Below, we provide our detailed responses to your comments.
>
> > **Restrictive Application Settings:** First, FourierSHAP can only applied when input spaces are binary. This means numerical features cannot be represented. This is a big limitation of the method, which obvious competitors like [1-4] do not have, which reduces the applicability of FourierSHAP in settings with numerical data. I am sure workarounds and fixes exist (binning and binarization), however this is left unexplored by the paper.
>
> We have addressed the binary assumption concern quite extensively in our answer to reviewer g97w. We ask you to kindly read this section and tell us what you think.
>
>
> Regarding competitors [1-4]; thank you for shedding light on these very interesting methods, which we will now discuss.
>
> Please correct us if we are wrong, but after going through [1] it seems this is a method that only applies to predictor models within the RKHS paradigm as Kernel regression, Kernel classification etc.  The models we have tested in our paper namely trees and neural networks do not fall under this category. [1] paper has no such experiments on these class of functions either and have a few experiments on synthetic data. We understand you are asking us to fit a an RKHS model to a MLP or a tree and then run these methods and compare to ours. Even though that sounds like an interesting thing to do, it lacks the thorough theoretical and empirical foundations we have presented for the case of Fourier representations. There are a few open questions here: 1- What kernel do we use for the case of trees and MLPS 2-Why is that kernel a useful function approximation for these classes of models
>
> While, we have provably and empirically shown that Fourier transform capture a compact representation for trees. And we have built upon previous thorough and justified empirical and theoretical results of why using Fourier transforms for MLPs make sense.
>
> The same reasoning applies for [2] which computes SHAP values efficiently when the predictor model is a (potentially multi-output) GP. Again trees and MLPs do not fall under this category. The paper has no experiments whatsoever trying to approximate MLPs or trees with GPs. And again the difference of our presentation with this paper is that we have justified the use of a Fourier representation for trees and MLPs thoroughly using our own derivations and also citing empirical and theoretical evidence from others on why this actually makes sense.
>
>
> [3] does not compute interventional SHAP values, but rather conditional SHAP values. In our paper we have not presented an algorithm for conditional SHAP values for trees and as mentioned explicitly in section 2.1 exclusively focused on the interventional setting.
>
> [4] however is a compelling relevant work that computes interventional SHAP values for piecewise linear regression trees. As normal decision trees are instances of piecewise linear regression trees, this work is an appropriate method to compare ours with. The paper also provides two new algorithms for computing interventional SHAP values for trees and show significant speedups over classic interventional TreeSHAP [*1]. We regret missing this work and are grateful to you for highlighting it. We took the time to include it into our baselines for the tree setting experiments and rerun all the experiments. We updated our Figure 3 in the paper and it now includes the fastest version of PLTreeSHAP introduced in [1].
>
> [5] we have included this as as baseline already (TreeShap) and we have orders of magnitude speedup over this method
>
> It was great going through these works, and adding the only direct competitor as a baseline into our tree setting experiments. We will definitely include all [1-4] in our related work. Again, thanks for mentioning them in your review.
>
> >Second, FourierSHAP seems to work only with very shallow neural networks (i.e. 3 layers with 300 neurons), which definitely is not realistic given modern architecture sizes.
>
> While we understand modern CNNs and Transformer architectures can be tens of layers, here in this paper we have only made claims about MLPs. Please again correct us if we are wrong, but practically speaking MLPs, rarely go beyond more than say 4,5 layers. Furthermore, our work relies on results from Spectral biases of such networks which still hold even as MLPs get deeper. The work of Yang and Salman shows the effects empirically and theoretically up to MLPs of 10 layers.

---

> ### Author Response · Authors · 2024-11-21
>
> > Usually, for XAI use cases testing with smaller networks is ok for model-agnostic methods, as the focus lies in showing that such methods can work with all sorts of models. However, given that FourierSHAP relies on finding a sparse Fourier representation of the underlying model, the aforementioned argument is invalid.
>
> We are not sure we understand why you are classifying Fourier SHAP as not being model agnostic. The only assumption that needs to be made in order to extract a Sparse Fourier approximation is query access to the model and nothing more. And given the Fourier representation we can compute SHAP values.
>
> The reason we have introduced trees and MLPs, is that these two classes of functions indeed have sparse Fourier approximations. This substantiates that fact that the class of Fourier sparse functions indeed can be quite expressive as it capture two widely used classes of predictor models as a subset. Please note again our method will apply to any black box method since it only assumes query access and no access to the structure of the MLP or the tree. Indeed for the MLP we only do queries to the MLP and using no other information whatsoever about its weight or its structure. Even though for the case of trees we have used the tree structure to extract the spare Fourier trasnsform we could have very well treated it like a black box and used the Fourier methodology to extract its representation. It would have taken longer but this would still work, and we did this in our black box experiments on two of our datasets, GB1 and avGFP.
>
> We think that inherent in any model agnostic method is some regularity assumption about the black box in order to show that the method works. For example LIME assumes the model is well approximated but a locally linear function. We think we have done an extensive work to show why Fourier SHAP is a very useful approximation method for model agnostic settings.
>
> > This Fourier representation will most definitely be drastically different from smaller sized networks. This needs to be theoretically or at least empirically explored. Otherwise, I am doubtful the results generalize to large scale networks
>
> The spectral bias theorem results still hold for deeper MLPs. This is proven by Yang and Salman and also shown empirically for networks of up to depth 10. I think the main intuition here is that the network learns a low degree function no matter the depth.
>
> >Lastly, computing ground-truth SVs with an approximation based method like KernelSHAP is an okay proxy, but in smaller feature settings like the Entacmea (13 features) dataset used in this paper or potentially synthetic settings, real ground truth values can be obtained through brute force or clever tricks with knowledge about the underlying data generating process / black box function.
>
> Regarding your concern about using KernelSHAP as ground truth; in our answer to reviewer g97w's Q3, we discussed this extensively, mentioning our toy tests with multiple predefined Fourier functions to make sure our formula for computing SHAP values off a sparse Fourier function works.
>
> In any case, as a proof of concept to help readers grasp the idea easier, we would be happy to compute SHAP values using the original SHAP formula and contrast it to KernelSHAP and ours, for Entacmaea.
>
> >**Presentation can be Refined:** The papers could be presented better. The work is more verbose than it needs to be. This starts in the abstract which is overly detailed and too long. The paper looses itself in discussing aspects that are in my opinion secondary and can be postponed to the appendix, like the parallelization and GPU acceleration. The paper contains multiple typos.
>
> Judging the breadth and depth of your knowledge showcased by your comments, we think that you are probably a more seasoned and experienced researcher in this field. The other 4 reviewers have all given a score of 3 for presentation and three of them have mentioned the presentation to be a strength. They seem to like the presentation and seem to appreciate the slower pace and verboseness of the topic. We think that it might be safer to keep it this way to make sure the work is accessible to a wider audience.
>
> > **Missing Limitations Section:** The paper does not contain a limitation section discussing FourierSHAP's drawbacks and avenues for interesting future work.
>
> We agree with you that a dedicated limitations and future works section will solidify the manuscript. We already discussed some of our ideas for extending this work in our response to reviewer g97w. We make sure to add this section to the paper if we get accepted.

---

> > ### Author Response · Authors · 2024-11-21
> >
> > > How does the Figure 1 look if the underlying model is a large neural network that potentially operates on text (text is also binary)?
> >
> > The text data, as noted, is usually inherently of binary/categorical nature.
> > While it is technically feasible to compute SHAP values on large neural networks trained on binary text, this is not widely used in practice. Text-based feature spaces like bag-of-words with realistic vocabulary sizes tend to have extremely high dimensions. In such cases, SHAP values can become difficult to interpret due to the sheer number of features and the challenge of identifying meaningful patterns in explanations. This practical limitation makes SHAP less commonly used for text compared to tabular data. We believe the datasets and models used in our experiments are more representative of current SHAP applications in the ML community.
> >
> > While large neural networks and binary text data might quantitatively amplify runtime or dimensionality challenges, it can technically be done in the scope of our proposed method.
> >
> > [*1] Lundberg et al. 2020; From local explanations to global understanding with explainable AI for trees: https://www.nature.com/articles/s42256-019-0138-9

---

> > > ### Comment · Reviewer_trxy · 2024-11-22
> > > **Response to Authors**
> > >
> > > Thank you to the authors for replying to my review. Please forgive me for maybe missing something in this response, but your replies (also to _g97w_) are quite extensive and **I want to focus on the main points here**:
> > >
> > > **First**, I do not really get your response to my question regarding the text-data and large transformer models. I do not really see why "SHAP [is] less commonly used for text", when it is a perfectly viable method to do so. Recently, I came across a NeurIPS paper from last year that does something like this in the experiments [1]. The SHAP library even has a whole section on this in their documentation [2]. A quick Google Scholar search also points to related papers here [3,4]. In fact, we can compute Shapley values for any set-valued function $\nu : \mathcal{P}(N) \rightarrow \mathbb{R}$, where $\nu$ could be any kind of model operating on some features like a text classifier/regressor. If you have a rather short sentence with a couple of words you can even compute the Shapley values exactly through brute force or you estimate them through any-model agnostic method like KernelSHAP (that's what they do in [1]). **My point with the question** was not in explaining some text classifier _because these models are so interesting to learn something about_, but rather to provide you with **a setting where FourierSHAP should be applicably** (all assumptions hold; i.e. binary data) **and** the model is not a _3-Layer MLP_ but a bigger neural network. You correct me that I assume FourierSHAP to be model-specific to Neural Networks or Trees by saying that FourierSHAP only requires _query access to the model and nothing more_. But then you say that _here in this paper we have only made claims about MLPs_ which is exactly the point **why I was raising the question regarding bigger models and how well FourierSHAP fares in these scenarios**. I would, of course, be also happy to see any other scenario that is not text involving an interesting black box model to analyze with FourierSHAP. Otherwise I still am doubtful about the true nature of FourierSHAP's _model-agnostic applicability_ and _performance_.
> > >
> > > **Second**, I very much appreciate the effort you took with _Interventional TreeSHAP_. However, now, I am a bit confused. You write in your response _in section 2.1 exclusively focused on the interventional setting_, which is true as you introduce the interventional baselines there. But why exactly does FourierSHAP only work with interventional feature removal? You did not say this in your manuscript. Appendix B.3 also just reiterates on the discussion regarding this topic in the XAI community but does not specify FourierSHAP's relation to this.
> > >
> > > **Third**, I mentioned RKHS-SHAP or GP-SHAP as important related work that, like your method, deals with kernel-based representations of models, which you are dealing with as well. You do not necessarily have to empirically compare the methods. I see your points you made in your response in that regard. Delineating FourierSHAP with those methods, discussing similarities and, importantly, the differences are in my opinion important. This would well fit into a dedicated related work section in your work (which likewise to a limitation section) is currently absent from the main body of your manuscript. This is why I criticize your presentation (and apparently do not agree with the remaining reviewers) in that you should rethink the structure of certain arguments and verbosity in favor of those **missing sections and important content** of your work.
> > >
> > > **Fourth**, after reading _g97w_'s review and your detailed response, I see that this seems to be an important limitation that should be adequately addressed. The minimum would be a dedicated limitation section. However, tackling these limitations to some extend can greatly improve the contribution of your work.
> > >
> > >
> > >
> > >
> > >
> > >
> > > [1] https://openreview.net/forum?id=IEMLNF4gK4&noteId=pL6cYtCk7N
> > >
> > > [2] https://shap.readthedocs.io/en/latest/text_examples.html
> > >
> > > [3] https://aclanthology.org/2021.hackashop-1.3/
> > >
> > > [4] http://proceedings.mlr.press/v119/sundararajan20b/sundararajan20b.pdf

---

> > > > ### Author Response · Authors · 2024-11-23
> > > >
> > > > Thank you for your quick response, it is very much appreciated. We apologize if the rebuttal seems verbose, it was not our intention to be overwhelming. You have raised a lot of concerns, some of which were quite fundamental. We spent a lot of time going over your response in detail trying to address each of your concerns the best we can. It was definitely our intention to be concise and to the point.
> > > >
> > > > **Q1**- It was not clear from your previous question what you exactly meant by binary text data and we now understand what you are referring to. Thank you for the interesting references and we agree that this sort of analysis on text data can be quite a nice application. However this does not fit the setting of our paper. Regarding [1], the NeurIPS paper, is based on Shapley **Interaction** values which seems more interesting and useful concept in language sentiment analysis.  Unfortunately, we do not have a proposed algorithm for **Interaction values**. In our case, if we did this experiment, we would simply be  looking at the output of a predictor and by removing and adding certain words and that might not be that interesting, even though that is the subject of paper [3]. In [1] they look at the adding and removal of words in the context of other words and their **interactions**.
> > > >
> > > > As we mentioned in the previous response, practically speaking, we think this algorithm can find applications mostly categorical datasets with discrete features. We have explained extensively, in answer to Q1 of reviewer g97w (which we referred you to), why we think this paper already provides value in the context of a variety of datasets, that contain only categorical features and how we think it should be extended to cover other tabular datasets that contain continuous features as well.
> > > >
> > > > That being said **there is nothing stopping us from running Fourier SHAP on a sentiment analysis dataset** like [3]. As we explained before this method is Model-Agnostic. Quoting reviewer:
> > > >
> > > > >You correct me that I assume FourierSHAP to be model-specific to Neural Networks or Trees by saying that FourierSHAP only requires query access to the model and nothing more. But then you say that here in this paper we have only made claims about MLPs which is exactly the point why I was raising the question regarding bigger models and how well FourierSHAP fares in these scenarios. I would, of course, be also happy to see any other scenario that is not text involving an interesting black box model to analyze with FourierSHAP. Otherwise I still am doubtful about the true nature of FourierSHAP's model-agnostic applicability and performance
> > > >
> > > > Fourier SHAP is model-agnostic method, i.e. it only assumes query access to the predictor. What we have gone ahead and done is that we have **guranteed** that for certain class of models namely trees and MLPs, and more generally speaking any model that can be represented by a Sparse Fourier approximation, the results produced by Fourier SHAP will perfectly coincicde with the ground truth i.e. the exponential formula. This does not mean it can not be applied to other classes of models. The fact that we have chosen neural nets and trees for our experiments is to showcase that our claims about the guarantees to be true. Furthermore they are the most commonly used models for datasets that we want to cover with this method, i.e. tabular datasets.
> > > >
> > > > Please note at the heart of of any model agnostic method, including kernel shap there is an approximation and this approximation can sometimes come with guarantees. In the case of kernel shap this appears as asymptotic statistical guarantees. Basically for a large enough number of sampled subsets kernel-shap recovers the ground truth shap values. In the worst case one has to sample all possible $2^n$ subsets. It is the same for our case: one can get better and better approximations of the black box by increasing the sparsity parameter. In the worst case one can get a perfect approximation using $2^n$ queries to the black box i.e. computing the whole fourier spectrum.
> > > >
> > > >
> > > > The authors of kernel shap, furthermore proved that for simple models such as linear predictors in the form of $f(x_1, \dots, x_n)=a_1 x_1+ \dots + a_n x_n$, kernel shap exactly recovers ground truth shap values . Please see section 4.2  of [1] for a list of specific models kernel shap is guaranteed to produce ground truth values for.

---

> ### Author Response · Authors · 2024-11-23
>
> The point being made here is that **just because the model agnostic method comes with guarantees for a  certain class of prediction models (like linear models in kernel-shap) does not imply it can not be used for other classes**. Kernel-Shap is still a model agnostic method because it only assumes query access to the black box. Going back to FourierShap, **the fact that we have provided some form of guarantees for MLPs and trees does not make the method model-based**. **We have presented a model-agnostic, query access, black box method that is guaranteed to produce ground truth shap values for two classes of widely used models namely MLPs and trees**. In our opinion, our contribution is important since for the case of MLPs, it is something that to our knowledge, no model-agnostic method has been able to do before. More precisely, **Except for FourierSHAP, there is no SHAP computation method that assumes only query access to the model i.e. is model agnostic, and is guaranteed to compute ground truth SHAP values for MLPs**.
>
> Just to drive the point home, we think the application of Fourier-Shap is just as justified as is kernelShap to the NLP datasets of [3] if not more, since it is guaranteed to work on a much more complicated class of prediction models than KernelSHAP has guarantees for. **We will try but not promise to run FourierSHAP on a language dataset before the end of the rebuttal period**. Please note, since FourierShap comes with guarantees for MLPs and trees we have focused a lot of effort in the paper portraying these guarantees to hold and we have focused the experiments on showing results in these classes of prediction models and chosen datasets that are relevant to these models.
>
> [1]: Lundberg, Scott M., and Su-In Lee. "A Unified Approach to Interpreting Model Predictions."
>
> Question 2:
> As mentioned before in Section 2.1 lines 137-144 we have clearly stated we **only** cover the interventional setting. Quoting those lines:
> >..., in this work, we focus on the SHAP values as defined in \textsc{KernelShap} introduced by [..] also known as  **Interventional**  [..] or Baseline  [..] SHAP, where the missing features are integrated out from the **marginal distribution** $p(x_{[n] \setminus S})$, as opposed to the conditional distribution $p(x_{[n] \setminus S} | x_S)$.
>
> The reason why it only covers this is because we do not have a way to tractably approximate the conditional distribution of a dataset. Capturing conditional distribution of a dataset is in general tricky and is even harder when one only has query access to the black box.
>
> In model-based methods like tree-shap, life is easier, as one can use the tree itself to approximate conditional distributions. We don't know if our method can cover this precise setting where the conditional is assumed to arise from the tree  as we have not thought about it. If the paper is accepted we will add this as a potential future work.
>
> Q3: We are glad you were convinced by our explanations here. As mentioned before, if the paper is accepted we will make sure to mention all of papers [1]-[4] and explain why they differ with our setting in the other work section of the paper in the appendix.  [4] was the only directly comparable paper and we have added it as a baseline as you suggested.
>
> Q4: Thank you for taking the time to read. As mentioned before, if accepted we will add a future work/limitation section including all the points mentioned there and the potential extension for the conditional tree setting (that we promised here in question 2).
>
> We thank you again for taking the time to review this work and adding your valuable comments.

---

> > ### Comment · Reviewer_trxy · 2024-11-29
> > **Maintaining my View of this Work**
> >
> > Thank you for the additional response. While I am happy about the theoretical contribution of this work, **my view and main concerns regarding this work remain unchanged**: What summarizes my main concern here the most is your sentence :
> >
> > > That being said **there is nothing stopping us from running Fourier SHAP on a sentiment analysis dataset** like [3]
> >
> > **Exactly!** But you are not doing this (or similar). I think this is an important thing to do for an honest empirical evaluation of your work to see that FourierSHAP **is indeed** a method that works well in amortizing SVs for any black box model. This is for what other amortization methods like FastSHAP have been presented. As long as the paper does not do this kind of evaluation, I am extremely doubtful about this methods capabilities to do so, which would (technicalities aside) make the method not so model-agnostic if the performance of FourierSHAP is too low for larger scale networks to be of any practical use.
> >
> > However, I do not say that if FourierSHAP is not able to handle large-scale networks, that FourierSHAP is not a good contribution and unworthy of acceptance at a top tier conference. In this scenario though, the paper should still do a better empirical evaluation concluding in identifying the areas where FourierSHAP actually shines and outperforms existing methods while still highlighting the areas where it performs poorly. Otherwise a lot of future work might be wasted finding this out. In this case of FourierSHAP under-performing, you may consider showcasing other interesting application areas where an amortized Shapley computation offers new research directions. Taking MLPs, for example, from the top of my head, I could see the field of reinforcement learning (especially reinforcement learning from human feedback) might be of interest, where often shallow models and MLPs are used like in the  DQN Paper [1] or in [2] and [3]. There is a very active research field focusing on explaining these models. Offering quick/instantaneous SV could be of great benefit and potentially enable new training/learning strategies as a whole. I don't know, however, if FourierSHAP's limitation of binary input features prevent this setting.
> >
> > This is why I think the work in its current form does not warrant acceptance and **should be rejected**. Conducting a robust evaluation should be done with care and would require an additional round of review.
> >
> > ### References
> > [1] https://arxiv.org/pdf/1312.5602
> >
> > [2] https://arxiv.org/pdf/1707.06347
> >
> > [3] https://papers.nips.cc/paper_files/paper/2017/hash/d5e2c0adad503c91f91df240d0cd4e49-Abstract.html

---

> ### Author Response · Authors · 2024-12-01
>
> Thanks for your response. While we appreciate the reviewer's intent to expand our work to other sub-fields of Machine Learning such as Reinforcement learning and NLP, we think this is not adding to the presentation of the paper.  In the previous response we have clarified how the current algorithm can be applied to query access black-box models of any sort including those that the reviewer wants.
>
> We believe the scope of the paper is quite clear and we do not plan to expand it any further. We believe the paper has a substantial amount of non-trivial material theoretically and empirically linking spectral biases of MLPs and trees to the task of computing SHAP values. Adding further fields of importance serves to distract the reader from the content that is meant to be presented.
>
> In short, we have presented a new and novel algorithm for computing amortized SHAP values for the setting of Tabular/Discrete data. Our algorithm is mathematically and empirically justified and is guaranteed to work for trees and MLPs in the query a.k.a black-box setting. We believe, the paper already has areas where it shines as showcased by our experiments where we have provided order of magnitude speedups on a variety of datasets and settings.
>
> We do not think it is required for us to include experiments from RL and LLMs as part of our work to introduce FourierSHAP.  As a matter of fact, none of the similar methods proposed by the reviewer, go beyond classic tasks and datasets to introduce their methods and shining points. They definitely did not provide any such experiments on language data, LLMs or Reinforcement learning.
>
> Since the reviewer thinks our work, with empirical studies on 4 datasets, lacks sufficient empirical evaluation to be published and should be clearly rejected, we have conducted a list of the baseline methods the reviewer suggested in his initial review and their empirical scope. These are all from top ML conferences as the reviewer has required. In addition to the reviewers suggested baselines, we have also included a few other well known methods in the XAI community that we have used as baselines. The following table shows what datasets were included in these works.
>
> | **Method**                                      | **Venue**                        | **Experimented Datasets**                                                                   |
> |-------------------------------------------------|----------------------------------|---------------------------------------------------------------------------------------------|
> | RKHS-SHAP [1]                                   | NeurIPS 2022                     | Diabetes                                                                                    |
> | GP-SHAP [2]                                     | NeurIPS 2022 (Spotlight)         | California, Breast cancer, Diabetes                                                         |
> | Linear TreeSHAP [3]                             | NeurIPS 2022                     | Adult, Conductor                                                                            |
> | Interventional TreeSHAP [4]                     | AAAI 2023                        | 7 Classic Tabular Datasets: pu_act, isolet, Ailerons, house_16H, sulfur, superconduct, year |
> | TreeSHAP [5]                                    | Nature Machine Intelligence 2020 | Three medical datasets: NHANES I mortality, Chronic Kidney Disease, Hospital Duration       |
> | Original TreeSHAP, KernelSHAP, and DeepLIFT [6] | NIPS 2017                        | An original user study, MNIST                                                               |
> | FastSHAP [7]                                    | ICLR 2022                        | News, Census, Bankruptcy, CIFAR10, Imagenette                                               |
> | Linear Regression SHAP [8]                      | AISTATS 2021                     | Census, Bank Marketing, German Credit, Breast cancer                                        |
>
> We hope this shows the paper is on par with other published work at top ML conferences in terms of empirical evaluation.
>
>
> ## References
>
> [1] RKHS-SHAP: https://openreview.net/pdf?id=gnc2VJHXmsG
>
> [2] GP-SHAP: https://openreview.net/forum?id=LAGxc2ybuH
>
> [3] Linear TreeSHAP: https://openreview.net/forum?id=OzbkiUo24g
>
> [4] Interventional TreeSHAP: https://ojs.aaai.org/index.php/AAAI/article/view/26322
>
> [5] TreeSHAP: https://www.nature.com/articles/s42256-019-0138-9
>
> [6] A Unified Approach to Interpreting Model Predictions: https://papers.nips.cc/paper_files/paper/2017/hash/8a20a8621978632d76c43dfd28b67767-Abstract.html
>
> [7] FastSHAP: https://openreview.net/forum?id=Zq2G_VTV53T
>
> [8] Linear Regression SHAP: https://proceedings.mlr.press/v130/covert21a.html

---

### Official Review · Reviewer_tLmD · 2024-11-06

**Soundness:** 3
**Presentation:** 3
**Contribution:** 3
**Rating:** 6
**Confidence:** 2

**Summary:**

The paper provides a principled approach for estimating Shapley (SHAP) values in the context of explainable machine learning. A closed-form expression for exact computation of the SHAP values in terms of the Fourier representation of the predictor function is obtained, which is used to obtain a tractable algorithm (polynomial asymptotic computational complexity) for estimating the SHAP values. Extensive experiments show that the proposed methods beats several recently proposed approaches for SHAP estimation in terms of the accuracy-efficiency trade-off.

**Strengths:**

- The paper is well-organized and easy to read with sufficient background on techniques used and prior methods.
- A new principled approach for estimating SHAP values is proposed based on a sparse Fourier representation of the prediction function
- The closed form expression in terms of the Fourier representation allows the approach to perform well with better Fourier approximation or when the sparsity assumption holds.
- Extensive experiments show benefit of using the proposed approach over previous methods in the literature.
- The approach can be parallelized making it potentially even faster in practice.

**Weaknesses:**

- It is not clear how the SHAP estimate degrades with the approximation error in sparse Fourier approximation.
- The initial step for Fourier approximation can be computationally expensive.

**Questions:**

- Past work has shown several optimizations are possible for Kernel SHAP making it faster. Are they applicable to your approach?

---

> ### Author Response · Authors · 2024-11-26
>
> We thank you for your thoughtful review and are delighted to hear that you found the script easy to follow and the literature review sufficient and helpful. We apologize for publishing of response to your review later than the others. Hope we can still make sure we could cover your points.
>
> > It is not clear how the SHAP estimate degrades with the approximation error in sparse Fourier approximation.
>
> General picture: This is a hard question to answer. The Fourier approximation error is global measure of how accurate the approximation is. Whereas for any query to be explained we are doing more local perturbations around that certain instance to be explained. Perhaps one can try to derive a bound by averaging all $2^n$ possible input instances to be explained and assuming a uniformly random background dataset. But we think a bound like that would not be of practical use as in practice one use background data-points and query points from the data distribution. So generally speaking without making very heavy assumptions about the data distribution (namely uniform) we do not see how one can practically derive a bound here.
>
> > The initial step for Fourier approximation can be computationally expensive.
>
> Yes, but as we have pointed out ourselves as well on lines 396-401 the sparse Fourier approximation is performed just once per model, and allows extremely cheap computation of SHAP values for any query inputs in an amortized fashion. We developed a JAX-based implementation of [1] that enables fast GPU-accelerated sparse Fourier approximation. As also mentioned in the Appendix F.1, using our implementation, we achieved high Fourier approximation accuracies in less than few minutes in all of our black-box experiments (Figure 1).
>
> On the other hand, training FastSHAP on multiple background datasets to run our experiments took up to multiple hours for some of our larger datasets. To give an example, it took us more than 8 hours to train FastSHAP on four background datasets to run our GB1 black-box experiments. This is while the approximated Fourier spectrum we achieved in 3 minutes already resulted in higher accuracies of SHAP values generated. Our method also does not need any such re-training with the change in the background dataset. This is orders of magnitude speedup in the first "pre-computation" step by comparison. We will include the extensive list of training times of all different versions of FastSHAP we trained for our experiments in Appendix F.1.3. This will highlight how cheaper overall our method is compared to similar methods that enable cheap amortized SHAP values computation.
>
> Furthermore, as it can also be seen in our Figure 1, sparse Fourier approximation is guaranteed to result in more and more accurate representations of the black-box function with the time spent. That is not necessarily the case when training a larger model for FastSHAP (this can be seen in our Figure 2). One can also take into account all the hyperparameter tuning and architecture engineering required to properly train a FastSHAP model (as can be seen in Appendix F.1.3), which is not needed in approximating sparse Fourier spectrum of a predictor.
>
> > Past work has shown several optimizations are possible for Kernel SHAP making it faster. Are they applicable to your approach?
>
> The optimizations in KenrelSHAP are mostly regarding smart ways of sampling coalition subsets. These results can not directly be used here as our method is inherently different. We believe the most promising optimizations for our method would likely lie in the initial sparse Fourier approximation step or in the parallel implementation of the second step.
>
> [1] Efficiently Learning Fourier Sparse Set Functions: https://papers.nips.cc/paper/9648-efficiently-learning-fourier-sparse-set-functions

---

> > ### Author Response · Authors · 2024-12-01
> > **Followup**
> >
> > Hello,
> > Just wanted to follow up to make sure we have addressed all of your questions. Not sure if you had the time to go through the new Appendix F1.3 that we added to the paper, specifically in response to your concern about the initial step being slow. There we added the only comparable baseline that we are aware of in the amortized setting, namely FastShap, and showed that we are quite a bit faster on our datasets compared to them in the first pre-computation step. More specifically we showed that, taking a sparse Fourier transform of the black-box takes considerably less time than training of a neural network to predict SHAP values.
> >
> > We hope with this addition, along with the above answers to your other questions we have addressed all your questions and concerns. If the reviewer thinks so too, we  would appreciate if you could raise your score.

---

### Author Response · Authors · 2024-11-21
**Updated Submission**

Dear Reviewers,

We have updated our submission to address some of your valuable comments. The main changes are:
1. Adding [1] as a new baseline in the tree setting (requested by reviewer trxy). See figure 3 of section 5 (experiments)
2. Correcting typos and refining some sentences for clarity (requested by reviewer 3voa, g97w)
3. Including more details in the proof of Lemma 1, to make it self-contained and easier to follow (requested by reviewer 3voa).

[1] Zern, A. and Broelemann, K. and Kasneci, G.; Interventional SHAP Values and Interaction Values for Piecewise Linear Regression Trees; Proceedings of the AAAI Conference on Artificial Intelligence, 2023.

We sincerely appreciate your time and thoughtful feedback.

Best regards,
Authors

---

### Author Response · Authors · 2024-11-23
**Update with Ground Truth SHAP values for the Entacmaea dataset - Figure 5 Added**

Dear Reviewers,

We have updated our submission to include the comparison requested between ground truth SHAP values, KernelSHAP outputs and FourierSHAP (ours) values. This was requested by "lacking empirical evaluation" of reviewer trxy and in Q3 of reviwerg97.  The results are in Appendix F.3 Figure 5.

We computed ground truth SHAP values using the original exponential SHAP formula for the Entacmaea. This is a 13-dimensional binary dataset and therefore small enough for us to apply the SHAP original exponential summation formula (Equation (1) in our paper. Furthermore it contains all possible  feature combinations in the dataset itself. We can see that both our results and KernelSHAP results perfectly coincide with ground truth SHAP values.

Note our results are extracted from the Fourier approximation of a neural network. Since in this case the neural network R2 score on the dataset is 1, it perfectly captures the dataset. We can see we are furthermore perfectly approximating the neural netowrk function with our Fourier expansion and hence computing precise SHAP values.

This is added to Appendix F.3, which was a dedicated section to discuss our process for making sure the KernelSHAP settings we used in our experiments ensure their convergence to ground truth values.  Hope it helps in reducing the potential confusion on using KernelSHAP values as ground truth in our black-box experiments.

Best regards,
Authors

---

### Author Response · Authors · 2024-11-28
**Update with adding "Limitations & Future Work" section and FastSHAP training times**

Dear reviewers,

We have updated our pdf submission with the following changes:

1- Added "Limitations & Future Work" section as Appendix G. We promised to add this section to reviewers g97w, trxy, and GoTf. There we discussed our ideas for two main directions for future work; extending the work to continuous features and computing conditional SHAP values.

2- Included training times of FastSHAP models we used in our black-box experiments as Table 1 in Appendix F.1.3. This was what we promised to reviewer tLmD, and highlights how our method is considerably faster than FastSHAP in the initial "pre-computation" step.

Both FourierShap (our method) and FastSHAP are amortized methods with a heavier pre-computation step that enables very cheap computation of SHAP values. For FastShap this pre-computation appears as training an MLP that directly predicts SHAP values and for us this appears as computing the Fourier transform of the predictor.

Best regards,
Authors

---

### Meta-Review · Area_Chair_nAH6 · 2024-12-17

**Metareview:**

I personally appreciate the topic of the paper, of computing SHAP values out of Fourier transforms. Given the span of the ratings, I did read the paper to make my own idea of its content and how the authors approached the rebuttal period, in particular in response to reviewer trxy.

In short, I believe it should have been a reasonably easy path forward to convince trxy, but for some reason the authors chose to stick to their guns during the whole rebuttal process, without giving ground on many clear questions of the reviewers, questions that would have been easy to answer (and this, after all, is the goal of the rebuttal process). Let me explain this:

* dealing with pseudo-boolean functions and the "restrictive application setting". This was an expected remark, and the authors make an extensive reply to this for g97w, split in two for trees and MLPs. But I believe the authors have completely missed a simple argument that would have justified the technique, if just for one of most prominent models currently used: trees -- and one most prominent kind of data: tabular data (only one occurrence in the submission !), which would also have been sufficient to consider the paper for acceptance. Indeed, they stick to the exhaustive binning of continuous features, which is okay but forgets the key argument of why trees do indeed work so well: only a subset of this expensive binning is sufficient to get good models. While this has lots of experimental support, the theory also comes in handy: the theory of boosting, which for decision trees proceed by assuming that *some* splits are good enough to beat random splitting (see the work of M. Kearns and Y. Mansour, STOC 96), clearly connects to the way learning *empirically* works. Algorithms like XGBoost (cited by the authors) have given up a bit of convergence rate to be applicable to large domains, but the authors could have stick to the algorithms first focusing on convergence rate (e.g. CART, C4.5/C5, etc) to develop a much more convincing argument of "what matters" in the size.

* experiments: I do not understand why the authors clearly devoted a lot of time summarizing existing papers (comment made Dec 1, 14:13), but did not choose instead to run experiments on a few more datasets, even of size compatible with the current implementation and hardware of their system, to content reviewers, instead of writing "there is nothing stopping us from running Fourier SHAP on a sentiment analysis dataset" (what is the purpose of a rebuttal period, then ?).

and there is more:

* when asked about a "Missing Limitations Section", instead of crafting one (easy !) during rebuttal time, the authors only promise it will be done in the future (again, what is the point, then, of having a rebuttal time ?). I am absolutely convinced that the authors could have softened the stance of trxy using just a few more experiments and highlights on a "competency map" of their approach (where it works well / does not work well). In fact, it is explicit from the reviewer: "However, I do not say that if FourierSHAP is not able to handle large-scale networks, that FourierSHAP is not a good contribution and unworthy of acceptance at a top tier conference. In this scenario though, the paper should still do a better empirical evaluation concluding in identifying the areas where FourierSHAP actually shines and outperforms existing methods while still highlighting the areas where it performs poorly". I really do not understand why the path chosen by the authors was to use rebuttal time to refrain from doing simple things -- again (see above) the whole lot of reviewers' arguments at rebuttal time on testing LLMs / deep nets could have been easily circumvented by polarizing the rebuttal on tabular domains, adding new experiments (maybe picking 1-2 Kaggle competitions, for the appeal), showing the usefulness of the approach on such (I am convinced it can be useful), dealing with continuous features in a simple way for the extension argument (see above), and leaving MLPs/LLMs either in a discussion section on extending the scope, or just in appendices.

In its current form (for the paper) and given the way the rebuttal period went, it is hard for me to accept the paper in its current form unfortunately. Reviewer trxy did engage very substantially in the discussion, offered various angles for discussion, and I believe the authors could have found, in the time allotted for rebuttal, simple ways to substantially soften the stance of the reviewer. As one last word of encouragement, I am absolutely confident there is a simple path forward that provides a very fine contribution to a top-notch ML conference. I strongly encourage the authors to consider it.

**Additional Comments On Reviewer Discussion:**

The author's response to the simplest arguments and requests from the reviewers (in particular on adding a few more experiments on simple domains / not LLMs and drawing a competency map of the approach) was unfortunately underwhelming. The authors chose to focus rebuttal time on not doing very expensive experiments (e.g. on LLMs, which is fine) while simple ones could have been done.

---

### Decision · Program_Chairs · 2025-01-22

Reject